# Uterine Commensal *Peptostreptococcus* Species Contribute to IDO1 Induction in Endometrial Cancer via Indoleacrylic Acid

**DOI:** 10.3390/biomedicines12030573

**Published:** 2024-03-04

**Authors:** Qiong Wang, Yaqiong Liu, Weiwei Chen, Sha Chen, Minting Su, Yanqin Zheng, Wenjie Liu, Li Li, Liang Zeng, Quan Shi, Juan He, Yuanmin Qian, Xingcui Xuan, Qirong Wen, Gendie E. Lash, Kun Shi

**Affiliations:** 1Department of Gynecology and Obstetrics, Guangzhou Women and Children’s Medical Center, Guangzhou Medical University, Guangzhou 510623, Chinachirainwan@gmail.com (Q.W.); 2Division of Uterine Vascular Biology, Guangzhou Institute of Pediatrics, Guangzhou Women and Children’s Medical Center, Guangzhou Medical University, Guangzhou 510623, China; gendie.lash@hotmail.com; 3Department of Obstetrics, Maternal and Child Health Hospital of Hubei Province, Wuhan 430070, China; chensha2723@163.com; 4Department of Pathology, Guangzhou Women and Children’s Medical Center, Guangzhou Medical University, Guangzhou 510623, China; 5The Joint Center for Infection and Immunity, Guangzhou Institute of Pediatrics, Guangzhou Women and Children’s Medical Center, Guangzhou Medical University, Guangzhou 510623, China; 6The Joint Center for Infection and Immunity, Institute Pasteur of Shanghai, Chinese Academy of Sciences, Shanghai 200031, China

**Keywords:** dysbiosis, indoleamine-2,3-dioxygenase, indoleacrylic acid, *Peptostreptococcus* species, type I endometrial cancer

## Abstract

Microbial dysbiosis has an increasingly appreciated impact on carcinogenesis, and the cervicovaginal microbiome plays a critical role in microenvironmental inflammation. Here, we investigated the involvement of the female genital tract *Peptostreptococcus* species in gynecological cancer via indoleacrylic acid (IAA). IAA production from *Peptostreptococcus* species and the effect of bacterial culture on tumor growth in vivo were examined. The impact of IAA on cytokine production and indoleamine-2,3-dioxygenase 1 (IDO1) expression in an endometrial cancer (EC) cell line, as well as their effect on T_reg_ and T_eff_ cells, and M1 and M2 macrophage populations were examined in EC patients and tumor-grafted mice. Clinically, *Peptostreptococcus* species abundance, IAA, and IDO1 expression were verified in EC patients. The results showed that IAA production was induced in the uteri of BALB/c nude mice by *Peptostreptococcus* species transplantation, and the intratumoral injection of a conditioned medium from *Peptostreptococcus* cultures into tumor-grafted mice promoted tumor growth. IL-10 expression was upregulated by IAA; IFN-γ expression was increased by IL-10. IFN-γ induced IDO1 expression in the EC cell line. The co-culture of IDO1-expressing EC cells with peripheral blood mononuclear cells upregulated the T_reg_ proportion and decreased the M1/M2 ratio. Clinically, *P. anaerobius* was more abundant amongst the uterine microbiota of EC patients than the control. The IAA, IDO1, and kynurenine/tryptophan ratios were all higher in EC tissue, and the M1/M2 ratio was lower. Our study sheds light on the link between IDO1 induction and uterine *Peptostreptococcus* dysbiosis and provides a potential rationale for the role of *Peptostreptococcus* species in immune tolerance induction in type I endometrial cancer.

## 1. Introduction

Endometrial cancer (EC) accounts for 4–8% of cancers in women; it is the 5th most common cancer in women worldwide and the 14th deadliest [1]. The rates of EC incidence and mortality continue to increase, even in premenopausal women and women younger than 40 years [1]. It is estimated that the incidence of EC will reach 13–42 per 100,000 by 2030 in the United States [1]. The human microbiome—which includes bacteria, viruses, and fungi—varies depending on demographic factors like age, race, and environment and clinical variables like disease progression and treatment [2]. Therefore, a number of studies have analyzed the link between the human microbiome, disease progression, and cancer therapy efficacy and identified several potential correlations relevant to the diagnosis, treatment, and prognosis of various cancers [3,4,5,6,7].

The cervicovaginal microbiome plays a key role in microenvironmental inflammation and infection [7,8]. In premenopausal and postmenopausal women, a healthy vaginal microbiome is primarily populated by *Lactobacillus* species, which produce lactic acid to help maintain the acidic environment and the low diversity of the microbiome, thereby protecting against the outgrowth of pathogenic microbial communities [9]. Estrogen therapy was proven to significantly increase vaginal lactobacillus levels in clinical trials [10], indicating the potential involvement of estrogen. Disturbances in the vaginal microbiome trigger a chronic inflammatory state in the upper genital tract, and this may contribute to the risk of pelvic inflammatory disease [11], spontaneous miscarriage [12], or gynecological cancers, such as uterine carcinogenesis [7]. The vaginal microbiome composition of women with gynecological cancer is consistent with that of women with symptomatic bacterial vaginosis [13], which indicates that bacterial vaginosis may be associated with a high risk of carcinogenesis. However, the role of the microbiome in the carcinogenesis or development of gynecological cancers is not completely understood. Further studies are required to determine the link between gynecological cancer biology and the microbial metabolome and multifaceted framework of microbial dysbiosis.

Tumor immune tolerance plays a crucial role in cancer recurrence and metastases. Dysbacteriosis is often linked to alterations in inflammatory response, which contribute to a plethora of diseases, such as lung cancer [14], colorectal cancer [15], and oral squamous cell carcinoma [16]. Wlodarska et al. reported that, in the gastrointestinal tract, commensal *Peptostreptococcus* species carrying the gene cluster *fldAIBC* produce a tryptophan metabolite, indoleacrylic acid (IAA) [17], which suppresses inflammatory responses by promoting IL-10 expression in macrophages. Moreover, *fldAIBC* expression is lower amongst the microbiome of inflammatory bowel disease patients. Interestingly, we recently reported that IAA is one of the most abundant metabolites in the sera of type I EC patients [18]. In addition, Chen et al. demonstrated the presence of commensal *Peptostreptococcus* species in the genital tracts of 110 healthy women of reproductive age, mainly in their endometria, cervical mucus drawn from the cervical canal, and posterior fornix [19]. Therefore, an overgrowth of *Peptostreptococcus* species in the endometrium could be a potential source of the high levels of IAA in EC patients. However, the effects of IAA on inflammation and immune responses during EC carcinogenesis are not known.

The CD4^+^CD25^+^CD127^(low/−)^ phenotype specifically identifies regulatory T cells (T_reg_) in human peripheral blood [20]. CD4^+^CD25^+^CD127^(low/−)^ T cells play immune-suppressive roles in acute pancreatitis [21], systemic lupus erythematosus, rheumatoid arthritis [22], chronic hepatitis B virus infection [23], and hepatitis B virus-associated hepatocellular carcinoma [24]. Indoleamine-2,3-dioxygenase (IDO) catabolizes tryptophan (Trp) via the kynurenine (Kyn) pathway, thereby promoting T_reg_ differentiation and inducing effector T cell (T_eff_) anergy [25,26]. In EC patients, a high IDO1 expression correlates with chemoresistance and low progression-free survival [27], but the mechanism of IDO1 induction is not clear. IDO1 can be induced in various human cancer cells or immune cells by IFN-γ stimulation [25,26,28]. IFN-γ expression can be induced by IL-10 in CD8^+^ T cells in skin cancer [29], suggesting the possible immunological regulation of IDO1 expression through IL-10 and IFN-γ. 

Here, to investigate the link between IDO1 induction and uterine *Peptostreptococcus* dysbiosis, we examined T_reg_ and T_eff_ populations and IDO1 expression in patients’ endometrial tissue. We investigated IAA production by *Peptostreptococcus* species and whether IAA potentiates IDO1 expression through the regulation of IL-10 and IFN-γ. We examined whether *Peptostreptococcus* species were involved in IDO1 induction through IAA. Finally, we determined whether M1 or M2 macrophages are the main sources of IL-10 after IAA stimulation. 

## 2. Materials and Methods

### 2.1. Patients

In total, 32 EC patients (29–73 years of age) diagnosed with type I endometrioid endometrial carcinoma by pathology based on FIGO staging guidelines were recruited from November 2017 to December 2019 in the Department of Obstetrics and Gynecology of Guangzhou Women and Children’s Medical Center, China. In total, 32 patients (31–70 years of age) diagnosed with endometrial hyperplasia (HP) and 32 benign patients (BN) (41–66 years of age) diagnosed as hysteromyoma or with a benign ovarian cyst were recruited from January to December 2019 in the Department of Obstetrics and Gynecology of Guangzhou Women and Children’s Medical Center, China. All the patients with any of the following criteria were excluded: (1) infections within the female genital tract, vaginal inflammation, endocrine or autoimmune disorders, or other chronic diseases (such as diabetes, hypertension); (2) a history of hormones, antibiotics, or vaginal medicine treatment in the last 6 months, or cervical treatment within a week; or (3) douche or sexual activity within 48 h. Age, body mass index (BMI), and menopausal status between the BN, HP, and EC groups showed no significant statistical difference; EC stages and grades were presented according to the FIGO 2009 Surgical Staging System for Endometrial Cancer; the types of HP and NC for the patients in this study cohort are described (Table 1). 

After EC, HP and BN patients were diagnosed through preoperative dilation and curettage, and their endometrial tissues and endometrial microbe specimens were collected under anesthesia intraoperatively in order to ensure fresh tissue collection; microbe samples from the cervix and posterior fornix were collected preoperatively. All the gynecological microbe samples were collected by rubbing dry sterile nylon flocked swabs on patients’ genital tracts, then transferring the swab heads into tubes with a sterile PBS solution, which were stored at −80 °C until required for examination. In total, 5 mL of peripheral blood from each patient (BN, HP, EC) was collected in an EDTA-coated tube for flow cytometry analyses and HPLC analyses.

### 2.2. Peptostreptococcus Species Abundance in Gynecological Microbiota

The abundance of 9 *Peptostreptococcus* species, including *P. anaerobius*, *P. russellii*, *P. stomatis*, *P. magnus*, *P. micros*, *P. asaccharolyticus*, *P. prevotii*, *P. tetradius*, and *P. productus*, was determined in the endometrium, cervix, and posterior fornix of all patients (EC, HP, and BN) by 16S rRNA gene examination. Bacterial genomic DNA extraction from all the gynecological microbe samples was performed with a Genomic DNA Purification Kit (Thermo Fisher SCIENTIFIC, Shanghai, China). 16S rRNA abundance of each *Peptostreptococcus* species was examined using real-time quantitative PCR with the following primers:

*P. anaerobius*:

F: 5′-ACGTGCTACAATGGGTGGTA-3′; R: 5′-CCTTCGACGACTTCCTCCTT-3′.


*P. russellii:*


F: 5′-TGAGATGACAGGTGGTGCAT-3′; R: 5′-ATTTGACGTCATCCCCACCT-3′.


*P. stomatis:*


F: 5′-GTAGTAAGCCGCCGAAACTG-3′; R: 5′-CTGTTTGCTACCCACGCTTT-3′.


*P. magnus:*


F: 5′-AGGTGGGGATGACGTCAAAT-3′; R: 5′-CGCGATTACTAGCAACTCCG-3′.


*P. micros:*


F: 5′-GGCAGCAGTGGGGAATATTG-3′; R: 5′-CATACGTATTACCGCGGCTG-3′.


*P. asaccharolyticus:*


F: 5′-GGCAGCAGTGGGGAATATTG-3′; R: 5′-CTTTACGCCCAGTGATTCCG-3′.


*P. prevotii:*


F: 5′-TACGGCGGGGTCTAGAGATA-3′; R: 5′-ATTTGACGTCATCCCCACCT-3′.


*P. tetradius:*


F: 5′-CTTGAGAGAGTGTACGGCCA-3′; R: 5′-CTTACGTATTACCGCGGCTG-3′.


*P. productus:*


F: 5′-TCCGGTGGTATCAGATGGAC-3′; R: 5′-CAATATTCCCCACTGCTGCC-3′.

Universal 16s rRNA primers:

27F: 5′-AGAGTTTGATCCTGGCTCAG-3′; 355R: 5′-GCTGCCTCCCGTAGGAGT-3′.

### 2.3. Reagents and Cell Lines

Primary antibodies, anti-IDO1, anti-FoxP3, anti-iNOS, anti-ARG1, anti-β-actin; Alexa Fluor-conjugated secondary antibodies, Goat Anti-Rabbit IgG H&L (Alexa Fluor^®^ 647 (Molecular Probes, Eugene, OR, USA)) and Goat Anti-Mouse IgG H&L (Alexa Fluor^®^ 488), and Hoechst 33342 were purchased from Abcam (Shanghai, China). FITC mouse anti-human CD3, PE mouse anti-human CD4, BV421 mouse anti-human CD25, AF647 mouse anti-human CD127, PE-CY7 mouse anti-human CD8, FITC rat anti-mouse CD3, PE-CY7 rat anti-mouse CD8, 7-AAD, FITC mouse anti-human CD45, PE-CY7 mouse anti-human CD68, PE mouse anti-human CD80, BV510 mouse anti-human CD86, AF647 mouse anti-human CD163, BV421 mouse anti-human CD206, Fc block, Fixation/Permeabilization Solution Kit, and FITC Annexin-V Apoptosis Detection Kit-I were bought from BD Biosciences (Shanghai, China). Recombinant human IL-10 and recombinant human IFN-γ were purchased from PeproTech (Suzhou, China). Trp and IAA were purchased from Sigma-Aldrich (Sigma-Aldrich, Shanghai, China). The DMEM medium and FBS were purchased from Thermo Fisher SCIENTIFIC (Shanghai, China). Human endometrial cancer cell lines, Ishikawa and HEC-1-B (Genechem, Shanghai, China) and the murine cervical cancer cell line U14 (Guangzhou Suyan Biotechnology, Guangzhou, China) were cultured in DMEM medium supplemented with 10% fetal bovine serum (FBS) and antibiotics (penicillin 100 U/mL, streptomycin 0.1 mg/mL, amphotericin B 0.25 μg/mL) and maintained at 37 °C in a humidified incubator with 5% CO_2_. All the cell lines were authenticated by short tandem repeat profiling and DNA sequencing. The shIDO1-EC cell line, IDO1^+^-EC cell line, and shVector-EC cell line were constructed as described previously [30]. Briefly, full-length human IDO1 cDNA (accession numbers NM_002164.6) were amplified in DH5α cells (Invitrogen, Carlsbad, CA, USA), cloned into lentivirus vectors LV011-pHBLV-CMV-MCS-3FLAG-EF1-T2A-Zsgreen-Puro (Hanbio Biotechnology, Shanghai, China) to construct an overexpression vector, and noted as IDO1^+^. Short hairpin RNAs (shRNAs) that selectively targeted (shIDO1-1: 5′-GGATGCATCACCATGGCATAT-3′), (shIDO1-2: 5′-GCCAAGAAATATTGCTGTTCC-3′), and (shIDO1-3: 5′-GGAGAATAAGACCTCTGAAGA-3′) were amplified and cloned into lentivirus vector pHB-U6-MCS-CMV-ZsGreen-PGK-Puro (Hanbio Biotechnology, Shanghai, China) to construct knockdown vectors noted as shIDO.

### 2.4. Bacterial Strains

*Peptostreptococcus* strains: *P. anaerobius* (CCUG 7835), and *P. russellii* (CCUG 58235), *P. stomatis* (CCUG 51858) were purchased from the Culture Collection University of Gothenburg (Göteborg, Sweden). Bacterial strains were grown in a Brain Heart Infusion (BHI) plus medium which was BHI (Becton Dickinson, Franklin Lakes, NJ, USA)-supplemented with 5% heat-inactivated FBS, a 1% vitamin K1-hemin solution (Millipore, Milwaukee, WI, USA), 1% Basal Medium Eagle vitamins (Sigma-Aldrich, Sanit Louis, MO, USA), 3 mM of D-(+)-cellubiose, 3 mM of D-(+)-maltose, 6 mM of D-(+)-fructose and 4 mM of L-cysteine. The medium was maintained at 37 °C in a tube with the medium surface blocked by Vaseline.

### 2.5. Animals

Experimental animal ethical approval was obtained from the Experimental Animal Ethics Committee of Guangzhou Medical University (accession number 2019-452). Mice were housed 5 mice/cage in an environmentally controlled room: temperature 21–22 °C; 12/12 light/dark cycle; fed daily with 4–8 g of standard mouse chow per mouse and water ad libitum. Female BALB/c nude mice (6–7 weeks old, 18–22 g) (Slac Laboratory Animal, Shanghai, China) were injected subcutaneously with Ishikawa cells (1 × 10^7^) in one front flank. Female C57BL/6 J mice (specific-pathogen-free (SPF)-grade, 18–22 g, 4–6 weeks old) (Slac Laboratory Animal, Shanghai, China) were injected subcutaneously with U14 cells (1 × 10^7^) in the front flank. When the tumor grew up to 100 mm^3^, the mice were randomized into treatment and control groups using a randomized block design based on tumor volumes (15 mice/group × 14 groups, 210 mice in total). Tumor volume (V) was expressed in mm^3^ using the formula: V = 0.5a × b^2^, where a and b are the long and the short diameters of the tumor, respectively. Mice were euthanized by cervical dislocation at the end of the experiment.

To in vivo examine IAA production by *Peptostreptococcus* species, three strains of *Peptostreptococcus* species (*P. anaerobius*, *P. russellii*, and *P. stomatis*) were inoculated into the uteruses of BALB/c nude mice (5 × 10^6^ CFU). Two months later, mice were euthanized, and the uteruses were collected for IAA examination. In bacteria-inoculated mice, a cell culture medium containing antibiotics was not used.

### 2.6. Trp, Kyn, IAA, IL-10, and IFN-γ Examination

Hyperplasia tissue from HP patients, EC, and adjacent tissue from EC patients were dissolved (1:5, *w*/*v*) into 0.4 M perchloric acid (PCA) (0.1% sodium metabisulfite, 0.05% EDTA), cut into small pieces (1 mm^3^), and then homogenized. The homogenate was centrifuged at 21,000× *g* for 10 min, and the collected supernatant was diluted (1:5, *w*/*v*) by 70% PCA and re-centrifuged at 21,000× *g* for 10 min. Then, the supernatant was collected for examination. 

The concentrations of Trp, Kyn, and IAA in the homogenate supernatant, sera, or cell culture media were analyzed by HPLC (Waters, Shanghai, China). In total, 20 μL of the sample was injected into a Symmetry C_18_ column (5 μm, 4.6 mm × 150 mm; Waters) through an autosampler. The mobile phase consisted of 50 mM of sodium acetate and 7.0% acetonitrile in ultrapure water (pH 6.20). The column temperature was set at 25 °C, and the flow rate was 0.4 mL/min. Trp was measured by UV absorption at 225 nm with a retention time of 8.4 min; Kyn was measured by UV absorption at 365 nm with a retention time of 4.0 min; and IAA was measured by UV absorption at 323 nm with a retention time of 9.8 min.

The concentrations of IL-10 and IFN-γ in the homogenate supernatant, sera, or cell culture media were examined by the Human IL-10 ELISA Kit (Abcam, Shanghai, China) and Human IFN-γ ELISA Kit (Abcam, Shanghai, China) according to the manufacturer’s instructions.

### 2.7. CD8^+^T cells and CD4^+^CD25^+^CD127^−^T Cell Proportions

A single-cell suspension, dissociated from tissues (HP, EC, and BN tissue), was prepared according to the following protocol. The tissues were washed 3 times with Hanks’ Balanced Salt Solution (Sigma-Aldrich, Shanghai, China), cut into small pieces (around 1 mm^3^), washed again, and then transferred into a sterile culture dish (60 mm). The tissues were dissociated by type I collagenase (1:7–1:10, *v*/*v*) (Sigma-Aldrich, Shanghai, China) at 37 °C for 20–40 min. Then, the tissues were transferred into a 50 mL centrifuge tube via filtering twice through a sterile nylon filter membrane (40 μM), and the cell suspension was available. After centrifugation at 250× *g* for 5 min, the cell pellet was collected, washed with Hanks’ solution, and centrifuged again. The cell pellet was re-suspended in a 2 mL RPMI-1640 medium, and cell density was calculated. For each tissue, a single-cell suspension with a total number (1 × 10^5^–1 × 10^8^) was required for the following flow cytometry analyses. Peripheral blood samples were treated via an erythrocyte lysing solution for 20–30 min (1:10, *v*/*v*) (KeyGEN BioTECH, Nanjing, China) before it was submitted to flow cytometry analyses.

Cells were stained with the following fluorescence-conjugated antibodies: FITC mouse anti-human CD3 (20 μL/sample), PE mouse anti-human CD4 (20 μL/sample), BV421 mouse anti-human CD25 (5 μL/sample), AF647 mouse anti-human CD127 (20 μL/sample), PE-CY7 mouse anti-human CD8 (5 μL/sample), and 7-AAD (2 μL/sample). Then, multi-color flow cytometry was performed by BD FACS Canto plus (Beckton-Dickinson, Sparks, MD, USA). The data were analyzed using FlowJo software (V10.0.7, TreeStar, Ashland, OR, USA). Gating strategies are shown in Appendix A; the isotype control test was performed, and data are shown in Appendix A.

### 2.8. CD68^+^CD80^+^CD86^+^ Cell and CD68^+^CD163^+^CD206^+^ Cell Proportions

A single-cell suspension prepared from tissue and blood samples was performed as described above. Then, cells were permeabilized by a fixation/permeabilization solution for 20 min at 4 °C, blocked by an Fc block for 30 min at 4 °C, and stained with the following fluor-conjugated antibodies: FITC mouse anti-human CD45 (20 μL/sample), PE-CY7 mouse anti-human CD68 (5 μL/sample), PE mouse anti-human CD80 (20 μL/sample), BV510 mouse anti-human CD86 (5 μL/sample), AF647 mouse anti-human CD163 (5 μL/sample), and BV421 mouse anti-human CD206 (5 μL/sample). Then, the cells were submitted to multi-color flow cytometry performed by BD FACS Canto Plus, and the data were analyzed using FlowJo software. The gating strategies are shown in Appendix A; the isotype control test was performed, and data are shown in Appendix A.

### 2.9. CD8^+^T Cell Proportions in Grafted Mice

A single-cell suspension was dissociated from the tissue and stained with FITC rat anti-mouse CD3 (50 μg/mL) and PE-CY7 rat anti-mouse CD8 (10 μg/mL). The cells then were submitted to flow cytometry, followed by data analysis using FlowJo software.

### 2.10. CD14^+^ PBMC and CD8^+^T Cell Isolation

In total, 5 mL of peripheral blood collected from BN, HP, and EC patients were mixed with 0.625 mL Optiprep^TM^ (Axis-Shield, Shanghai, China). A total of 0.5 mL of PBS was carefully dropped to the top, the mixed solution was centrifuged at 1500× *g* for 30 min, and PBMCs were collected with a Pasteur pipette beneath the white liquid. The PBMCs were diluted with sterile saline by double volume and centrifuged at 500× *g* for 10 min.

CD14^+^ PBMCs and CD8^+^T cells were enriched from PBMCs by a human MC CD14 Monocyte Cocktail (Miltenyi Biotec, Bergisch Gladbach, Germany) and human MACSxpress Whole Blood CD8 T Cell Isolation Kit (Miltenyi Biotec), respectively. CD8^+^T Cell proliferation was stimulated by PHA (5 μg/mL) for 48 h before the exam.

### 2.11. Differentiation of M1 and M2 Macrophages

M1 and M2 macrophage differentiation was performed, as previously described. Briefly, CD14^+^ PBMCs were treated with M-CSF (10 ng/mL) for 3 days, followed by detaching the cells, resuspending them with a fresh medium and culturing them with M-CSF (10 ng/mL) for 3 days. On days 6–7, the cells were differentiated in M0 macrophages. For M1 differentiation, macrophages were treated with LPS (10 ng/mL) and IFN-γ (5 U/mL) for 3 days, followed by a changing medium and culturing for 1 day. For M2 differentiation, macrophages were treated with IL-4 (20 ng/mL) for 3 days, followed by a changing medium, and culturing for 1 day.

### 2.12. Cell Apoptosis Assay 

Cell apoptosis was examined by Annexin-V/PI staining and analyzed by flow cytometry (Ex/Em = 488 nm/525 nm for Annexin-V detection and Ex/Em = 488 nm/610 nm for PI detection).

### 2.13. Cell Proliferation Assay

Cell proliferation was examined by the CellTrace carboxyfluorescein succinimidyl ester (CFSE) staining assay and analyzed by flow cytometry (Ex/Em = 492 nm/517 nm).

### 2.14. Cell Viability Assay

Cell viability was examined by the alamarBlue staining assay and analyzed by spectrophotometry at wavelengths 570 and 600 nm.

### 2.15. Immunofluorescence Staining

Cells were fixed on slides with 4% paraformaldehyde in PBS (pH 7.4) for 10 min at room temperature, incubated for 10 min with PBS containing 0.2% Triton X-100 for permeabilization, and then washed 3 times (5 min for each wash). Then, the cells were incubated in a blocking solution (1% BSA, 22.52 mg/mL glycine in PBST) for 30 min, followed by a primary antibody in 1% BSA in PBST in a humidified chamber overnight at 4 °C, including rabbit anti-IDO1 (1:500) and mouse anti-β-actin (5 μg/mL). The slides were washed 3 times in PBS (5 min for each wash), then incubated in secondary antibodies in 1% BSA for 1 h at room temperature in the dark, including Goat Anti-Rabbit IgG H&L (Alexa Fluor^®^ 647) and Goat Anti-Mouse IgG H&L (Alexa Fluor^®^ 488). The slides were washed 3 times in PBS in the dark (5 min for each wash), then incubated in 0.5 μg/mL of Hoechst for 3 min and rinsed in PBS. A mounted coverslip with a drop in the mounting medium sealed the coverslip with nail polish and collected the images with a Leica DMi8 inverted fluorescent microscope (Leica MICROSYSTEMS, Wetzlar, Germany). 

### 2.16. IHC Staining

All the tissues were formaldehyde-fixed, embedded in paraffin, and sectioned. Paraffin sections were baked in an oven at 60 °C for 4 h and then dewaxed with two changes in xylene for 10 min each, followed by hydration in graded ethanol for 5 min each. For antigen retrieval, the slides were immersed in a citric acid retrieval solution, heated in a microwave, and cooled down at room temperature. Endogenous peroxidase activity was quenched by 3% H_2_O_2_ for 15 min. After blocking nonspecific binding, the sections were incubated with a primary antibody at 4 °C overnight, including IDO1 (3 μg/mL), FoxP3 (10 μg/mL), iNOS (1:100), and ARG1 (1:100). Next, the sections were incubated with a biotinylated secondary antibody followed by incubation with the streptavidin-peroxidase conjugate and color development with DAB-H_2_O_2_. A negative control was included by replacing the primary antibody with PBS.

### 2.17. Real-Time Quantitative PCR

All the gynecological microbe samples were centrifuged at 8000× *g* for 10 min, and the supernatant was discarded. Total RNA was extracted from the microbe samples by a TRIzol Reagent (Thermo Fisher SCIENTIFIC, Shanghai, China) and reverse-transcribed by a PrimeScript RT Master Mix (Takara, Dalian, China). The primer pairs for the *fldAIBC* gene cluster were designed as shown below:

*fldAIBC*: F: 5′-ATGAACGATAAGTGTGCCGC-3′; R: 5′-GCAAGTCCCGCTACTCTACT-3′.

Data were analyzed and exported with the value of the threshold cycle (Ct), the differences in Ct values (ΔCt) between the test locus and the control locus (ACTB), and the comparative Ct (ΔΔCt) for the calculation of the difference between samples with fold change.

### 2.18. Statistics

The results obtained from clinical examinations, as well as the data obtained from in vitro and in vivo experiments, were analyzed by one-way ANOVA followed by Turkey’s multiple comparisons test; a survival curve comparison was analyzed by the Gehan–Breslow–Wilcoxon test (GraphPad Prism 6, GraphPad, La Jolla, CA, USA). Data were presented as the mean ± SD, and *p* < 0.05 was considered statistically significant.

## 3. Results

### 3.1. Type I EC Patients Had a Higher Proportion of CD4^+^CD25^+^CD127^−^ T Cells and Greater IDO1 Expression Than Normal Controls

In this study, EC patients had a higher proportion of CD4^+^CD25^+^CD127^−^ T cells in their peripheral blood (Figure 1A) and endometrial tissue (Figure 1B) than benign tissues (BN), and a lower proportion of CD8^+^ T cells in their endometrial tissue than BN or endometrial hyperplasia patients (HP) (Figure 1B). Immunohistochemical staining showed that IDO1 was significantly more highly expressed in EC tissue than in HP or BN tissue (Figure 1C). There were no differences in the serum Kyn/Trp ratios of BN, HP, or EC patients, but we observed a higher ratio in EC endometrial tissue than in BN or HP tissue (Figure 1D).

### 3.2. IAA Promotes IL-10 Secretion by Macrophages and IFN-γ Expression by CD8^+^ T Cells Which Contributes to IDO1 Induction

According to our studies, IAA is differentially expressed in EC patients [17], which shows highly expressed IDO1 in endometrial tissue (Figure 1C). Previous studies showed that IAA stimulates IL-10 production in the macrophages [16], and IL-10 promotes IFN-γ production in CD8^+^ T cells [28]. Therefore, we investigated if IAA-IL10-IFNγ contributes to IDO1 induction downstream of those effects and benefits immune tolerance. IAA-induced IL-10 production by peripheral blood-derived macrophages (Figure 2A). In turn, IL-10 promoted IFN-γ secretion by CD8^+^ T cells (Figure 2B). IFN-γ treatment induced IDO1 in Ishikawa endometrial adenocarcinoma cells. We observed an increase in CD8^+^ T cell apoptosis when Ishikawa cells were treated with IFN-γ (0.2 ng/mL) for 24 h and then co-cultured with CD8^+^ T cells for 24 h (Figure 2C).

In an in vivo assay, we injected Ishikawa cells (1 × 10^7^) subcutaneously into the front flank of BALB/c nude mice, followed by the intratumoral injection of IFN-γ (10 ng/kg). The IFN-γ-treated mice had higher IDO1 expression in tumoral tissue (Figure 2D) and larger tumors (Figure 2E) than vehicle-treated mice. We also injected the murine cervical cancer cell line U14 into the front flank of C57BL/6 mice, followed by the intratumoral injection of IFN-γ (10 ng/kg). The IFN-γ-treated mice showed increased FoxP3 positive staining in the grafted tumor tissue (Figure 2F) with fewer CD8^+^ T cells (Figure 2G) than in vehicle-treated mice. 

To further investigate the role of IAA in IDO1 induction through IL-10 and IFN-γ, experiments were performed, as depicted in Figure 2H. CM2, the conditional medium from the CD8^+^T cell culture in the IAA-treated group, promoted IDO1 mRNA and protein expression in EC cells compared with CM1 or CM3 (Figure 2I). CM2 also caused the increase in IDO1 concentration in tumor culture media (a tumor conditional medium (TCM), Figure 2J). TCM1-3 was then used to treat CD8^+^T cells (which were untreated previously) in order to examine TCMs’ effects. It was shown that TCM2 reduced Trp concentration (Figure 2K) and increased the [Kyn]/[Trp] ratio (Figure 2L) in CD8^+^T culture media and attenuated CD8^+^T cell viability (Figure 2M) compared with TCM1 or TCM3. For the in vivo assay performed in C57BL/6 mice (Figure 2H), the CM2-injected tumor graft showed evident tumor growth (Figure 2N), IDO1 expression in tumor tissue (Figure 2O), a lower relative Trp concentration (Figure 2P), higher relative [Kyn]/[Trp] ratio (Figure 2Q), as well as lower relative CD8^+^T proportions in tumor-infiltrated lymphocytes (TIL) (Figure 2R), compared with the tumor graft treated by CM1 or CM3. Here, relative values indicate the value in tumor foci (TF) divided by the value in peripheral blood (PB) or the value in tumor-infiltrated lymphocytes (TIL) divided by the value in peripheral blood lymphocytes (PBL), considering their individual difference in vivo. 

### 3.3. IL-10 and IFN-γ Play Roles in IAA-Induced IDO1 Expression 

Here, the IL-10 or IFN-γ antibody was applied to examine the roles of IL-10 and IFN-γ in IDO1 expression. First, IFN-γ production was promoted in CD8^+^T cell media by IAA-treated macrophage media and evidently inhibited by the IL-10 antibody (MΦ0–MΦ1000, indicating the macrophages treated by IAA 0–1000 μM) (Figure 3A). Then, the experiments were performed as depicted in panels B and C (Figure 3). Compared with TC1 EC cells (no treatment), TC2 cells (with IL-10 (0.5 ng) in the CD8^+^T culture) showed IDO1 expression, but their expression was blocked by the IFN-γ antibody (0.5 μg) (TC3 EC cells). IFN-γ (0.2 ng) stimulated IDO1 expression in EC cells (TC4), and the expression was not blocked in EC cells (TC5) treated by the IL-10 antibody (0.5 μg). IAA (50μM) induced IDO1 expression (TC6), which was blocked by IL-10 antibody-treated CD8^+^T cells (TC7) or in IFN-γ antibody-treated EC cells (TC9). But IDO1 expression was not blocked in IL-10 antibody-treated EC cells (TC10).

To further examine the impact of tumor conditional media with IDO1 on the viability and function of CD8^+^T cells (Figure 3D), tumor conditional media, TCM1, TCM2, TCM4, TCM6, TCM7, and TCM9, as well as CD8^+^T cells without previous treatment were used. The IDO1 expression in these tumor cells (TC1, TC2, TC4, TC6, TC7, TC9) (Figure 3E) and the concentration of IDO1 in TCMs (Figure 3F) were examined. TCM2, TCM4 or TCM6 showed IDO1 production in media compared with TCM1, TCM7 or TCM9. After being treated by TCMs, the CD8^+^T cell medium (TM2M, TM4M or TM6M) showed a lower Trp concentration and higher [Kyn]/[Trp] ratio compared with TM1M, TM7M, or TM9M (Figure 3G,H). Moreover, cell viability or proliferation was attenuated in TCM-treated CD8^+^T cells (TM2T, TM4T, or TM6M) compared to that in TM1T, TM7T, or TM9T (Figure 3I,J). TM2T, TM4T, or TM6T increased the apoptotic ratio compared with other TCM-treated CD8^+^T cells (Figure 3K).

For the in vivo examination performed with C57BL/6 mice, as depicted in Figure 3L, TCM1, TCM6, TCM7, and TCM9 were used to examine the impact of TCMs. Compared with tumor grafts (TMM1, TMM7, or TMM9), TMM6 showed faster tumor growth (Figure 3M), evident IDO1 expression (Figure 3N), a lower relative Trp concentration (Figure 3O), higher relative [Kyn]/[Trp] ratio (Figure 3P), as well as a lower CD8^+^T proportion (Figure 3Q). 

Clinically, EC patients had higher serum and endometrial tissue levels of IAA (Figure 3R), IL-10 (Figure 3S), and IFN-γ (Figure 3T) than BN, and their tissue IL-10 and IFN-γ levels were higher than those in HP.

### 3.4. Peptostreptococcus Species Produces IAA

IAA is produced as a metabolite of Trp by the gut microflora. Given that the expression of the *fldAIBC* gene cluster is associated with IAA production [17], the female genital tract contains commensal *Peptostreptococcus* species [19], and pathogenic microbes potentially contribute to EC development [31]; we examined the microbiota of the posterior fornix, cervix, and endometrium. We evaluated the abundance of 9 *Peptostreptococcus* species: *P. anaerobius* (PA), *P. russellii* (PR), *P. stomatis* (PS), *P. magnus*, *P. micros*, *P. asaccharolyticus*, *P. prevotii*, *P. tetradius*, and *P. productus*. EC patients had a higher abundance of PA amongst the microbiota of the endometrium (Figure 4A), cervix (Figure 4B) and posterior fornix (Figure 4C) than BN or HP. EC patients also had a higher abundance of *P. micros* in the cervical microbiome than BN or HP patients (Figure 4B), and a higher PR abundance in the posterior fornix microbiome than HP patients (Figure 4C). The abundances of the other *Peptostreptococcus* species did not differ between the three groups at any of the sites tested. EC patients showed significantly high PA(+) cases in endometrial or posterior fornix specimens, compared with BN patients (Figure 4D).

In addition, we compared *fldAIBC* expression in the microbiome of the posterior fornix, cervix, or endometrium of BN, HP, and EC patients. EC patients had a higher endometrial microbiome expression of *fldAIBC* than BN or HP patients and a higher expression in the cervical microbiome than BN (Figure 4E). There were no significant differences in *fldAIBC* expression in the posterior fornix microbiomes of EC patients, HP, or BN; all specimens from the endometria, cervix, and posterior fornix of BN, HP or EC patients showed *fldAIBC(+)* (Figure 4F). In addition, we examined the in vitro and in vivo IAA production of PA and PR, which carry the complete *fldAIBC* gene cluster, and PS, which has an incomplete *fldAIBC* gene cluster. We observed higher IAA levels in supernatants from PA or PR cultures than in PS culture supernatants (Figure 4G). BALB/c nude mice transplanted with PA or PR had higher IAA levels in the uteri or peripheral blood than PS-transplanted mice (Figure 4G).

### 3.5. Peptostreptococcus Species Contributes to IDO1 Induction and Immune Tolerance via IAA 

In order to examine whether *Peptostreptococcus* species could contribute to IDO1 induction in EC cells through macrophages and CD8^+^ T cells, we designed the in vitro experiment shown in Figure 5A. We cultured PA, PR, and PS for 3 days and measured the levels of IAA in the culture supernatants. PA- and PR-conditioned media had higher levels of IAA than the PS-conditioned medium (panel A). We stimulated macrophages with conditioned media for 24 h and collected the resultant macrophage-conditioned media. The PA-stimulated macrophage-conditioned medium had the highest IL-10 concentration of all the macrophage-conditioned media (panel A, MPA). We then stimulated CD8^+^ T cells with the macrophage-conditioned media for 24 h and collected the CD8^+^ T cell-conditioned media. The PA-stimulated macrophage-conditioned medium upregulated IFN-γ expression in CD8^+^ T cells (panel A, TPA). The PS+IAA (50 μM; PSI)-stimulated macrophage-conditioned medium also upregulated IFN-γ expression in CD8^+^ T cells to a greater extent than the PS-stimulated macrophage-conditioned medium (panel A, TPSI vs. TPS). Next, we examined the effect of CD8^+^ T cell-conditioned media on IDO1 expression by Ishikawa cells. The medium collected from PA-treated macrophage-stimulated CD8^+^ T cell cultures induced high IDO1 expression compared to the media collected from PR- or PS-treated macrophage-stimulated CD8^+^ T cell cultures (panel A, ECPA). In addition, the medium collected from PS+IAA-treated macrophage-stimulated CD8^+^ T cell cultures induced IDO1 expression compared with medium collected from PS-treated macrophage-stimulated CD8^+^ T cell cultures (panel A, ECPSI).

The experimental design of the in vivo experiments performed in C57BL/6 mice is shown in Figure 5B. We injected U14 cells into the murine front flank. When the tumor measured 100 mm^3^, the mice were grouped and treated with a vehicle or PA-conditioned, PR-conditioned, or PS-conditioned medium for 2 months. PA-treated mice had greater tumor volumes than non-treated mice (*p* < 0.05) or PS-conditioned medium-treated mice (*p* = 0.06) (panel B). PA-treated mice had a lower proportion of CD8^+^ T cells in their grafted tumors than PS-conditioned medium-treated or non-treated mice (panel B). Their tumor tissues also had higher IDO1 and FoxP3 expression (panel B) than tumor tissues from PS-conditioned medium-treated or non-treated mice.

PA is the main strain of *Peptostreptococcus* species in the female genital tract, and it showed strong IAA production and an evident effect on IDO1 induction in EC cells based on the results above; therefore, experiments were further performed in vitro (Figure 5C) and in vivo (Figure 5D) in order to examine whether PA has an effect on immune tolerance and whether IDO1 expression was involved. The TPA CD8^+^T medium in the PA group and TPS CD8^+^T medium in the PS group (panel A) were used (with TPS as the negative control) in order to evaluate the effect of PA. 

First, IDO1 expression was examined. Compared with TPS-treated WT EC cells and its TCM (TWS), TPA-treated WT cells showed IDO1 expression and IDO1 concentration in TCM (TWA). Compared with shVector EC cells and its TCM (TVA), IDO1 expression was upregulated in IDO1^+^ EC cells (*Ido1* overexpression), and the IDO1 concentration was high in TCM (TOA) but significantly impaired in shIDO1 cells (TNA) (panel C). 

The TCMs were used to treat CD8^+^T cells in order to examine their impact on them (depicted in panel C). Compared with TWSM (TWS treated-CD8^+^T medium), the Trp concentration in TWAM (TWA treated-CD8^+^T medium) was decreased, and the [Kyn]/[Trp] ratio was increased (panel C). Compared with TVAM (TVA treated-CD8^+^T medium), the Trp concentration was significantly low in TOAM (TOA treated-CD8^+^T medium), and the [Kyn]/[Trp] ratio was high while they were reversed in TNAM (TNA treated-CD8^+^T medium) (panel C).

Cell proliferation in all TPS-treated EC TCM-treated CD8^+^T cells did not alter significantly (panel C). In TPA-treated EC media-treated CD8^+^T cells (TWAT, TVAT, TNAT, or TOAT), we observed a significant difference in cell proliferation as well as apoptosis (panel C). Compared with TVAT (CD8^+^T treated by shVector-EC TCM), cell proliferation was low in TOAT (CD8^+^T treated by IDO1^+^-EC TCM) but was high in TNAT (CD8^+^T treated by shIDO1-EC TCM) (panel C). Moreover, compared with TWST (CD8^+^T treated by TPS-treated WT-EC TCM), cell apoptosis increased in TWAT (CD8^+^T treated by TPA-treated WT-EC TCM). Compared with TVAT, apoptosis was increased in TOAT but reduced in TNAT (panel C).

For in vivo experiments performed with C57BL/6 mice (as depicted in Figure 5D), compared to WT cells treated by TPS, WT cells treated by TPA showed fast tumor growth, evident IDO1 expression, reduced relative Trp concentration, an increased relative [Kyn]/[Trp] ratio, and decreased CD8^+^T proportion in TIL (panel D). Compared with shVector cells, IDO1^+^ cells showed more progressive tumor growth, strong IDO1 expression, a low relative Trp concentration, a high relative [Kyn]/[Trp] ratio, and a low CD8^+^T proportion in TIL (panel D); however, in shIDO1 cells, these phenomena in IDO1^+^ cells were reversed (panel D).

### 3.6. Transcriptional Factors p-NF-κB, p-STAT1 and IRF1 Were Involved in IAA-Induced IDO1 Expression in EC Cells and Immune Tolerance

It has been reported that the transcriptional factors p-NF-κB and/or p-STAT1 can be activated by IFN-γ and promote IRF1 expression, which promotes IDO1 expression in IFN-γ-treated human tumor cells [24,25,27]. In this study, p-NF-κB, p-STAT1 and IRF1 expressions were examined in EC cells (TC1, TC2, TC4, TC6, TC7, TC9, as illustrated in Figure 3C): p-NF-κB, p-STAT1, and IRF1 were expressed in TC2, TC4 and TC6, but were reduced in TC7 and TC9 (Figure 6A–C). Therefore, whether p-NF-κB, p-STAT1 and IRF1 were involved in IAA-induced IDO1 expression and IDO1’s impact on CD8^+^T cells was further investigated. In the experiments depicted in Figure 6D, compared with SVC cells (shVector EC cells), IDO1 expression was evidently inhibited in VPC (shVector EC cells with PDTC (NF-κB inhibitor) treatment) or STC (shSTAT1 EC cells); however, in TPC (shSTAT1 EC cells with PDTC treatment) or IFC (shIRF1 EC cells), it was completely inhibited. 

Then, TCMs were collected to treat CD8^+^T cells (depicted in Figure 6E). First, the IDO1 concentration in TCM was examined. Compared with SVM (TCM of SVC), the IDO1 concentration was significantly attenuated in VPM (TCM of VPC) or STM (TCM of STC) and even lower in TPM (TCM of TPC) or IFM (TCM of IFC) (Figure 6F). Then, CD8^+^T cells and media were collected and examined after TCM treatment. Compared with SVMM (SVM-treated CD8^+^T media), Trp concentrations in other CD8^+^T media (VPMM, STMM, TPMM, or IFMM) were higher; their [Kyn]/[Trp] ratios significantly decreased (Figure 6G,H). These TCM-treated CD8^+^T cells are as follows: VPMT (VPM treated), STMT (STM treated), TPMT (TPM-treated), and IFMT (IFM-treated), and they also presented high cell viability (Figure 6I), proliferation ratio (Figure 6J), and low apoptosis (Figure 6K) compared with SVMT (SVM-treated).

For in vivo experiments performed with C57BL/6 mice, 5 cell lines, shVector cells, shVector cells with PDTC treatment, shSTAT1 cells, shSTAT1 cells with PDTC treatment, and shIRF1 cells were subcutaneously inoculated in mice, followed by the intratumoral injection of the CD8^+^T conditional medium for 2 months and then an examination by tumor and blood (Figure 6L). Compared with SVCT (SVC tumor), tumor growth was attenuated in VPCT (VPC tumor), STCT (STC tumor), TPCT (TPC tumor), and IFCT (IFC tumor) (Figure 6M); IDO1 expression was significantly reduced (Figure 6N), relative Trp content was high (Figure 6O), the relative [Kyn]/[Trp] ratio was low (Figure 6P), and CD8^+^T proportion was upregulated (Figure 6Q). 

### 3.7. Peptostreptococcus Species Stimulated IL-10 Production in M2, but Not in M1 Macrophages

To determine if M1 or M2 macrophages produced IL-10 after *Peptostreptococcus* species stimulation, we treated M1 and M2 macrophages with bacterial supernatants collected from PA, PR, and PS, as well as negative control (NC) and PS+IAA (50 μM; PI) cultures (Figure 7A). PA- or PI-conditioned medium-treated M2 macrophages had higher IL-10 expression than similarly treated M1 macrophages (Figure 7B). We then collected the M1 and M2 macrophage-conditioned medium to treat CD8^+^ T cells. After 24 h, the M2 macrophage-conditioned medium induced the upregulation of IFN-γ expression in CD8^+^ T cells (Figure 7C), whereas the M1 macrophage-conditioned medium did not. Next, we collected the treated CD8^+^ T cell-conditioned medium to treat EC cells. The M2-conditioned medium-treated CD8^+^ T cell-conditioned medium induced IDO1 expression in Ishikawa cells (PAE2, PIE2), whereas the M1-conditioned medium-treated CD8^+^ T cell-conditioned medium did not.

Then, we co-cultured the Ishikawa cells treated with the conditioned media with CD8^+^ T cells for 24 h to investigate their effects on the survival of CD8^+^ T cells (Figure 7D). Compared to the Ishikawa cells treated with an M1-conditioned medium-treated CD8^+^ T cell-conditioned medium, the cells treated with the M2-conditioned medium-treated CD8^+^ T cell-conditioned medium increased the apoptosis of CD8^+^ T cells (Figure 7D).

To further investigate the effects of EC cells on T_reg_, CD8^+^ T cells, M1 macrophages, and M2 macrophages, we collected EC cells treated with CD8^+^ T cell-conditioned media and co-cultured with peripheral blood mononuclear cells (PBMCs) for 72 h (Figure 7E). We examined the proportions of CD4^+^CD25^+^CD127^−^ T cells and CD8^+^ T cells and the ratio of M1/M2 macrophages. EC cells stimulated with an M2-conditioned medium-treated CD8^+^ T cell-conditioned medium induced a higher proportion of CD4^+^CD25^+^CD127^−^ T cells (Figure 7F, PA2 vs. PA1, PI2 vs. PI1), a lower proportion of CD8^+^ T cells (Figure 7G, PA2 vs. PA1, PR2 vs. PR1), and a lower M1/M2 ratio in PBMCs (Figure 7H, PA2 vs. PA1) than EC cells stimulated with M1-conditioned medium-treated CD8^+^ T cell-conditioned medium.

### 3.8. Clinical Correlation

Clinically, we observed no differences in the M1/M2 ratios of the PBMCs of BN, HP, or EC patients, although the proportion of M2 macrophages was higher in the EC patients and HP than in the BN (Figure 8A). By contrast, EC patients had a lower M1/M2 ratio in their endometrial tissue than BN or HP (Figure 8B). In addition, in EC endometrial tissue, the proportion of M1 macrophages was lower, and the proportion of M2 macrophages was higher than in BN endometrial tissue (Figure 8B). We examined the expression of iNOS and ARG1 via immunohistochemical staining. We observed lower iNOS expression in EC endometrial tissue, including epithelial cells and mesenchymal cells, compared to BN tissue (Figure 8C). ARG1 expression was higher in EC tissue, including in neoplastic cells and mesenchymal cells, than in BN or HP tissue (Figure 8D).

Clinically, p-NF-κB, p-STAT1, and IRF1 expressions were examined in benign and cancer tissue. Compared with benign endometrial tissues, EC tissues presented significantly high expressions of p-NF-κB (Figure 8E), p-STAT1 (Figure 8F) and IRF1 (Figure 8G). Moreover, linear fitting degrees between the *fldAIBC* expression level and each of the following clinical values were examined (Figure 8H): the IAA, IL-10, IFN-γ, Trp content, and [Kyn]/[Trp] ratio in endometrium (ET), H-scores of p-NF-κB, p-STAT1, IRF1 and IDO1, as well as CD8^+^T proportions in endomtrial lymphocytes (ETL). 

## 4. Discussion

Understanding the molecular mechanism of tumor immune tolerance is a key issue in cancer biology. IDO is one of the major enzymes used by tumor cells to induce immune tolerance in the microenvironment; a high IDO expression correlates with poor prognosis in EC patients [27]. However, the regulation of IDO expression during EC development is still unclear. Our results suggest that commensal PA in the uterine microbiota may contribute to IDO1’s induction via the production of IAA. IAA promotes IL-10 production by macrophages in endometrial tissue, which, in turn, stimulates IFN-γ production by CD8^+^ T cells. IFN-γ production consequently stimulates IDO1 expression in EC cells, thereby accelerating Trp catabolism and Kyn production. The Trp starvation, as well as Kyn-induced T_reg_ differentiation, might interfere with the proliferation and cytotoxic function of CD8^+^ T cells, which potentially creates a local immune-tolerant microenvironment in EC tumor foci (Figure 9). 

Cancer is generally regarded as a disease caused by alterations in host genetics and environmental factors [30]; microorganisms are implicated in about 20% of human malignancies [32,33,34]. Microbes in mucosal microenvironments may contribute to the communication between epithelial cancer cells and immune cells, similar to aerodigestive tract cancer or urogenital tract cancer; in addition, intratumoral microbes may play a role in tumor growth and spread [35]. It is estimated that cancer-associated microbes designated as carcinogenic to humans comprise only 10 of approximately 3.7 × 10^31^ microbes on Earth [32]. Interestingly, compared with carcinogenic viruses that can integrate oncogenes into host genomes and drive carcinogenesis—such as the human papillomavirus (HPV), human hepatitis virus, and *Epstein–Barr* virus [32]—carcinogenic bacteria that trigger transformation events in host cells (such as *Helicobacter pylori* [35]) are rarely reported. Walther-António et al. reported that the microbiome compositions of the vagina, cervix, fallopian tubes, and ovaries all correlated with EC, which had a structural microbiome shift that distinguished it from benign conditions [31]. They concluded that *Atopobium vaginae* and *Porphyromonas* species in the gynecological tract, together with a high vaginal pH, were linked to EC progression, potentially indicating a role for the microbiome in EC etiology. Recently, several authors reviewed relevant studies, and it was suggested that the colonization of cervicovaginal microbes such as *Peptostreptococcus* species are significantly augmented through HPV infection and thus drive cervical carcinoma [36]. In the current study, we demonstrated that *Peptostreptococcus* species in the female genital tract may contribute to IDO1 induction in type I EC. This finding will inform future studies as to whether *P. anaerobius* is carcinogenic or contributes to immune tolerance induction or how the *P. anaerobius* population is enriched in the uterine microbiota of EC patients. These future studies are necessary to explore the potential pathophysiological role of uterine commensal bacteria in the context of gynecological carcinogenesis. Based on our results, we conclude that the dysbacteriosis of the uterine microbiota may correlate with the carcinogenesis of type I EC.

A growing body of evidence indicates that simple metabolites produced by commensal microbiota might play pathophysiological roles in human disease, which suggests that microbial dysbiosis could be, either directly or indirectly, pathogenic [37]. IAA is one of the Trp metabolites produced by the human microbiota. Wlodarska et al. reported that IAA production by *P. russellii* plays a key anti-inflammatory role by stimulating IL-10 expression in macrophages [17]. We confirmed that PBMC-derived macrophages and, in particular, M2 macrophages, upregulated IL-10 expression after IAA stimulation. In addition, EC patients had a lower endometrial M1/M2 ratio than HP patients or BN. However, there was no difference in the PBMC M1/M2 ratios of EC, HP, or BN patients. This finding indicates that the M2 macrophage population is upregulated in EC foci but not in the peripheral blood. The EC endometrial tissue also had lower iNOS and higher ARG1 expression than HP or BN tissue, confirming the lower M1/M2 ratio in EC foci.

M2 macrophages physiologically function as suppressors of Th1 cytokine-mediated inflammation. In the tumor microenvironment, they are regarded as a pro-tumour subpopulation of tumor-associated macrophages (TAMs). An overabundance of M2 TAMs may compromise the immune surveillance of cancer foci due to their production of anti-inflammatory cytokines and inhibition of antigen presentation and T cell proliferation [38,39]. In vivo therapy with anti-programmed cell death-1 Ab decreased ARG1^+^ TAM populations [39], which is consistent with another study on the TAM-mediated inhibition of tumor immunity via the expression of PD-1 [38]. We found that PA-conditioned medium-treated M2 macrophages induced IDO1 expression in EC cells and that IDO1 expression in EC cells stimulated the M2 differentiation of co-cultured PBMCs. Therefore, by producing IAA, PA communicates with M2 macrophages to promote their expansion or survival in tumor foci and drive IDO1 expression. 

Several questions have been raised by these results. It is unclear why IFN-γ, a well-known anti-tumor Th1 cytokine, was induced by treating CD8^+^ T cells with the M2-conditioned medium. Furthermore, it is intriguing that IFN-γ promoted IDO1 expression but not cancer cell apoptosis. It is possible that treatment with the M2-conditioned medium led to the production of sublethal IFN-γ levels by CD8^+^ T cells, which could indicate that CD8^+^ T cells promote IDO1 induction under specific circumstances.

Recent studies have shown that IDO1 expression positively correlates with CD8^+^ T cell populations amongst tumor-infiltrating lymphocytes in rectal carcinoma [40], hepatocellular carcinoma [41], and early-stage cervical cancer [42]. A statistically significant positive correlation was also observed between the transcript levels of IDO1 and IFN-γ in 144 cervical cancer samples [42]. In the margins of cervical tumors, where the tumors are surrounded by immune cells, tumor cells express IDO1; this is regarded as a consequence of T cell infiltration and local IFN-γ induction [42]. In addition, IDO1 is constitutively expressed in hepatocellular tumor foci [41]. Therefore, CD8^+^ T cell-derived IFN-γ could contribute to IDO1 expression in cancer cells. In our study, the CD8^+^ T cell-conditioned medium with a high IFN-γ concentration induced IDO1 expression in EC cells, unlike the medium with a low IFN-γ concentration. The co-culture of IDO1-expressing EC cells with CD8^+^ T cells caused apoptosis in CD8^+^ T cells. 

IFN-γ-induced IDO1 expression causes tryptophan starvation and kynurenine accumulation, which could have an extensive impact on tumor environment immune cells. In this study, the CD8^+^T cell population and apoptosis were examined. Natural killer (NK) cells were not examined in the present study but may also be influenced by IDO-caused tryptophan starvation. It has been reported that l-kynurenine-treated NK cells have impaired cytotoxicity, which suggests that NK cells may be suppressed in IDO-expressing conditions [43].

A study performed with melanoma, breast carcinoma, and colon carcinoma cells showed that the systemic degradation of Kyn reversed the effects of IDO1 in the tumor microenvironment and increased the tumor-infiltration and proliferation of CD8^+^ T cells [44], suggesting that IDO1 attenuates CD8^+^ T cell function and viability. We found that the serum Kyn/Trp ratio did not differ amongst EC patients, HP, or BN, but the Kyn/Trp ratio and IDO1 expression were highest in EC endometrial tissue. In addition, the proportion of CD8^+^ T cells amongst endometrium-infiltrating lymphocytes was lower in EC patients than in HP or BN patients. Taken together, our findings indicate that increased numbers of M2 TAMs in the tumor foci of EC patients increase IL-10 expression, which stimulates IFN-γ release from CD8^+^ T cells, leading to IDO1 expression by EC cells and the attenuation of tumor-infiltrating CD8^+^ T cell responses. In fact, the effect of IAA injection alone into the U14-grafted tumor in C57BL/6 mice could not be compared with the effect of a bacterial media injection, such as the PA injection. The bacterial medium of PA, PR, or PS was composed of several other factors besides IAA, and those factors could play a role in the recruitment of immune cells into the tumor site in C57BL/6 mice, while only injecting IAA might not. Accordingly, it was possible for the PA, PR, or PS medium to stimulate the production of IL-10 in macrophages and IFN-γ in CD8^+^T cells in the tumor site. But, for the IAA injection alone, the immune cells were rare in the U14-grafted tumor site, and it was difficult to induce IL-10 and IFN-γ production. Therefore, there are other factors within the bacterial conditional medium required for immune cell infiltration that plays a key role in IAA’s effect on inducing IDO1 expression.

After co-culture with IDO1-expressing EC cells, PBMCs had a higher proportion of T_reg_ and a lower M1/M2 ratio than PBMCs co-cultured with non-IDO1-expressing EC cells. Based on previous reports [26], the IDO1 product Kyn may promote T_reg_ differentiation. However, the role of T_reg_ in the differentiation of M2 macrophages in this setting is unclear. 

We may conclude here that PA dysbiosis causes IAA upregulation, which promotes IDO1 expression and Kyn production by regulating macrophages and CD8^+^ T cells with potential effects on immune surveillance in tumor foci. This work highlights the possible causal relationship between microbial dysbiosis and carcinogenesis and suggests its potential role in EC diagnosis, such as serving as an adjuvant diagnostic method together with imaging, curettage, etc. In the following study, we plan to recruit a large cohort to investigate the possibility of using PA abundance as an adjuvant diagnostic method. PA abundance in EC patients with different stages, prognostic, and recurrence conditions is another point to consider; we need to examine the difference in PA abundance and the correlation between PA abundance and EC stage, prognostics, and recurrence. If so, correcting dysbacteriosis in the tumor microenvironment could be associated with a better prognosis, and early intervention might ameliorate the therapeutic regimen clinically. 

In this work, the murine cervical cancer cell line U14 was used for inoculation into C57BL/6 mice since the xenografted tumor was not available through inoculating human endometrial cancer cell lines in C57BL/6 mice, and the murine endometrial cancer cell line was not available as well. In addition, in the tumor-grafted mouse model, we did not examine the levels of different populations of macrophages, which is perhaps a limitation of the study. Nevertheless, in the clinical endometrial tissues examined, we did find an alteration in the ratio of M1 to M2 macrophages in EC patients compared with HP or BN patients, which may support our hypothesis.

## Figures and Tables

**Figure 1 biomedicines-12-00573-f001:**
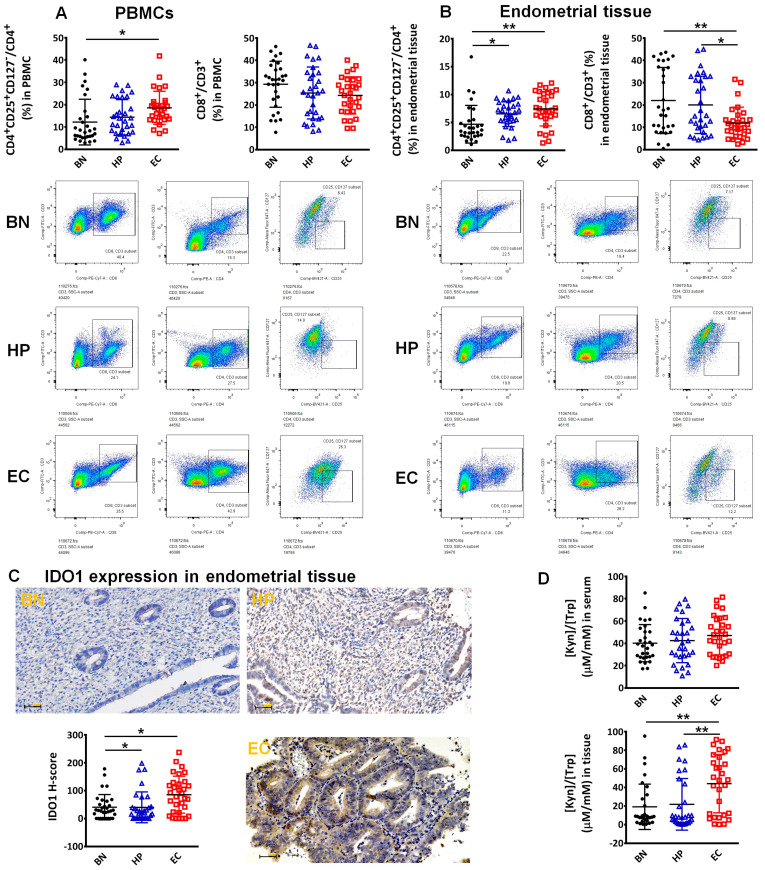
Proportion of CD4^+^CD25^+^CD127^−^T cells and CD8^+^T cells in PBMCs of BN, HP, and EC patients (**A**). Proportion of CD4^+^CD25^+^CD127^−^T cells and CD8^+^T cells in endometrial tissue of BN, HP, and EC patients (**B**). (**A**,**B**) Examined by flow cytometry. IDO1 expression in the endometrial tissue of BN, HP, and EC patients (**C**). Scale bars represent 50 μm. Kyn/Trp ratio in serum or endometrial tissue of BN, HP, and EC patients (**D**). “*” indicates *p* < 0.05; “**” indicates *p* < 0.01.

**Figure 2 biomedicines-12-00573-f002:**
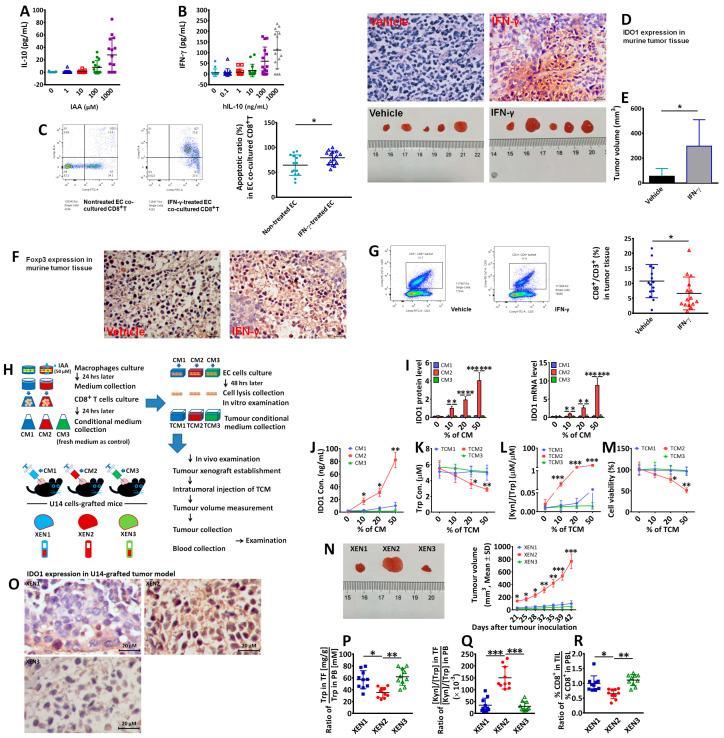
IL-10 expression in IAA-treated macrophages (0–1000 μM) (**A**). IFN-γ expression in hIL-10-treated CD8^+^T cells (0–1000 ng/mL) (**B**). Apoptosis of CD8^+^T cells co-cultured with IFN-γ-treated (0.2 ng/mL) or untreated EC cells examined (**C**). IDO1 expression in grafted tumor treated with intratumoral IFN-γ (10 ng/kg, 3 times per week for 2 months) or vehicle (**D**); tumor volume (**E**), FoxP3 expression (**F**) and CD8^+^T cells proportion (**G**) in grafted tumor (each group n = 15). In vitro and in vivo experimental procedures (**H**) (each group n = 10). IDO1 expression in CM-treated EC cells (**I**) and IDO1 concentration in tumor cells media (TCM) (**J**). TCMs were used to treat CD8^+^T cells (0–50% in CD8^+^T medium). Trp concentration (**K**) and [Kyn]/[Trp] ratio (**L**) in TCM-treated CD8^+^T media, as well as CD8^+^T cell viability (**M**). For in vivo experiments, CD8^+^T cell conditional media (CM1-3) were used to treat tumor grafts in C57BL/6 mice by intratumoral injection (3 times per week), and tumor volume was measured (**N**). Two months later, tumors and blood were collected to analyze IDO1 expression (**O**), relative Trp content (**P**), and [Kyn]/[Trp] ratio (**Q**), as well as relative proportions of CD8^+^T cells (**R**). “*” indicates *p* < 0.05; “**” indicates *p* < 0.01; “***” indicates *p* < 0.001.

**Figure 3 biomedicines-12-00573-f003:**
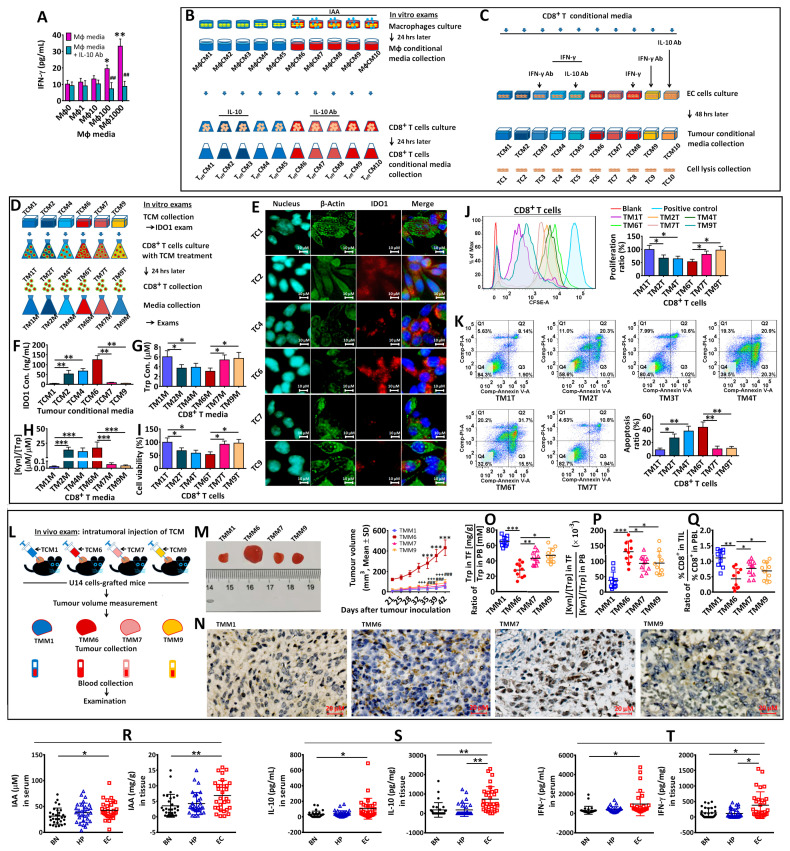
IFN-γ production by CD8^+^T cells after treatment with the media of IAA-treated macrophages (IAA: 0–1000 μM) (**A**). The roles of IL-10 and IFN-γ in IAA-induced IDO1 expression via EC cells were investigated in vitro (**B**). The IDO1 expressions by EC cells (TC1-10) were examined (**C**). Then, EC TCMs (TCM1,2,4,6,7,9) were used to treat CD8^+^T cells (**D**). The IDO1 expression by those EC cells was confirmed (**E**). The IDO1 concentration in their TCMs was examined (**F**). The Trp concentration (**G**) and [Kyn]/[Trp] ratio (**H**) in the media of TCM-treated CD8^+^T cells, as well as cell viability (**I**), proliferation (**J**), and apoptosis (**K**) in these cells were examined. For in vivo exams, U14 cells were inoculated into C57BL/6 mice, and their tumor grafts were treated with different TCMs (**L**) (each group n = 10). The difference in tumor growth was analyzed (**M**) (“***” indicates the difference between TMM6 and TMM1; “###” indicates the difference between TMM7 and TMM6; and “+++” indicates the difference between TMM9 and TMM6). Two months later, tumor grafts and blood were collected for the analyses of the IDO1 expression (**N**), relative Trp content (**O**), and [Kyn]/[Trp] ratio (**P**), as well as the relative proportion of CD8^+^T cells (**Q**). Clinically, IAA (**R**), IL-10 (**S**), and IFN-γ (**T**) concentrations in the sera and endometrial tissues of BN, HP, and EC patients were examined. “*” indicates *p* < 0.05; “**” or “##” indicates *p* < 0.01; “***”, “###” or “+++” indicates *p* < 0.001.

**Figure 4 biomedicines-12-00573-f004:**
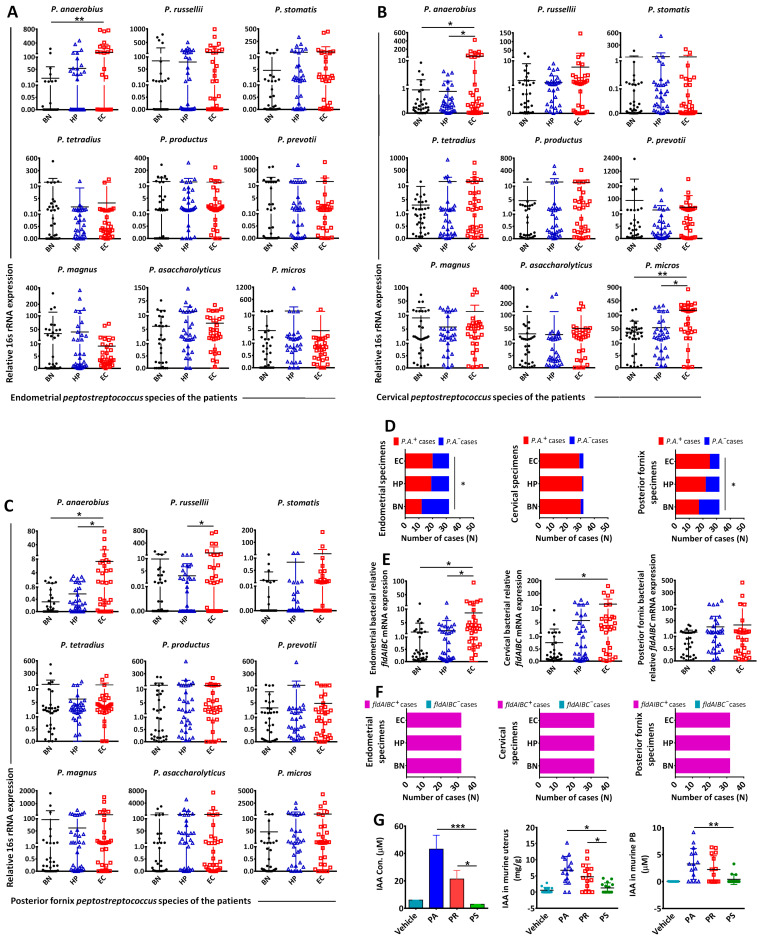
Examination of the abundance of 9 *Peptostreptococcus* species in the gynecological microbiota of BN, HP, and EC patients. *Peptostreptococcus* species included *PA*, *PR*, *PS*, *P. magnus*, *P. micros*, *P. asaccharolyticus*, *P. prevotii*, *P. tetradius*, and *P. productus*. Gynecological microbiota included endometrial microbiota (**A**), cervical microbiota (**B**), and posterior fornix microbiota (**C**). PA^(+)^ and PA^(−)^ cases in endometria, cervix, or posterior fornix were examined (**D**). Relative gene cluster *fldAIBC* expression in the endometrial, cervical, and posterior fornix microbiota of BN, HP, and EC patients (**E**), as well as *fldAIBC^(+)^* and *fldAIBC*^(−)^ cases of those microbiota (**F**). IAA production in *P. anaerobius* (PA), *P. russellii* (PR), or *P. stomatis* (PS)-culturing medium as well as in bacteria (PA, PR, or PS)-transplanted murine uteri and peripheral blood (**G**). “*” indicates *p* < 0.05; “**” indicates *p* < 0.01; “***” indicates *p* < 0.001.

**Figure 5 biomedicines-12-00573-f005:**
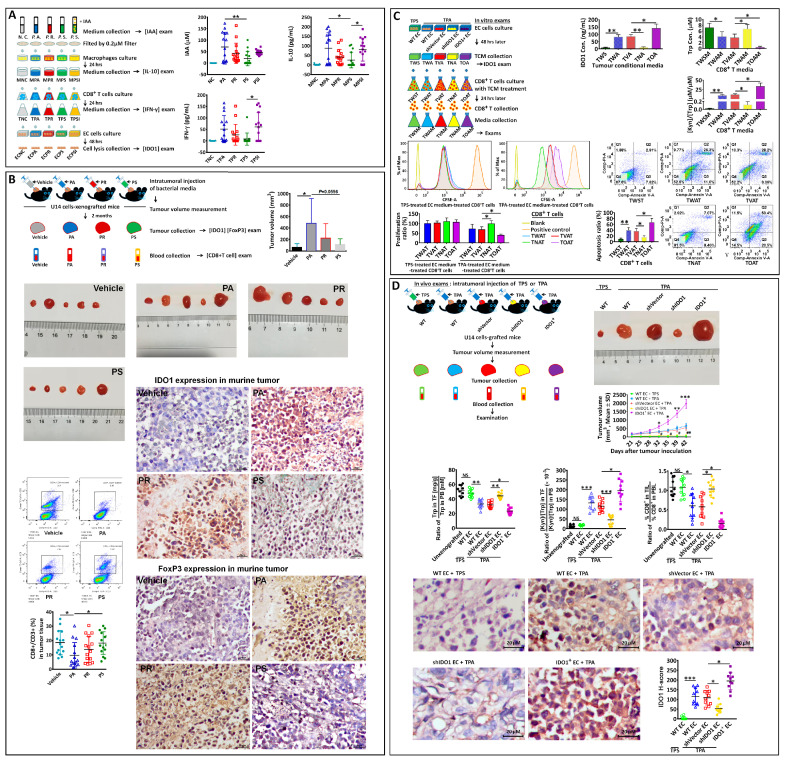
The scheme of in vitro examinations of the contribution of *Peptostreptococcus* species to IDO1 induction in EC cells (**A**). The examination of IAA, IL-10, IFN-γ, and IDO1 (panel (**A**)). The scheme of in vivo experiments of *Peptostreptococcus* species contribution to IDO1 and FoxP3 expression in tumor grafts in C57BL/6 mice (**B**) (each group n = 15). The examination of tumor volume, CD8^+^T cell proportion (panel (**B**)), as well as IDO1 and FoxP3 expressions in grafts. TPA (CD8^+^T conditional medium)-treated shIDO1 EC cells and IDO1^+^ EC cells were used to examine the role of IDO1 in PA’s impact on the tumor microenvironment, and the experiments are depicted in panel (**C**). The IDO1 expression by those EC cells and IDO1 concentration in their TCMs were examined (panel (**C**)). The Trp concentration and [Kyn]/[Trp] ratio in the media of TCM-treated CD8^+^T cells, their proliferation, and apoptosis were examined (panel (**C**)). In vivo exams are depicted in panel (**D**), including tumor growth (“*” indicates the difference between IDO1^+^ and shVector cells; “#” indicates the difference between shIDO1 and shVector cells), IDO1 expression, the relative Trp content, [Kyn]/[Trp] ratio, and relative proportions of CD8^+^T cells (each group n = 10). “*” or “#” indicates *p* < 0.05; “**” or “##” indicates *p* < 0.01; “***” indicates *p* < 0.001.

**Figure 6 biomedicines-12-00573-f006:**
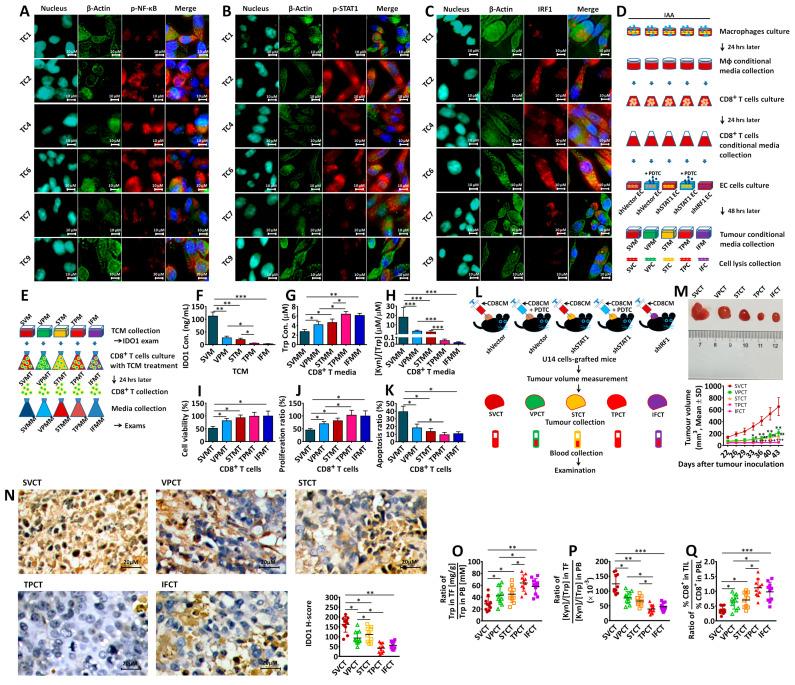
EC cells (TC1, TC2, TC4, TC6, TC7, TC9, as illustrated in Figure 3C) were used to examine p-NF-κB, p-STAT1, and IRF1 expressions (**A**–**C**). shVector EC cells, shVector EC cells with PDTC treatment (25 μM), shSTAT1 EC cells, shSTAT1 EC cells with PDTC treatment (25 μM), and shIRF1 EC cells were used to investigate the roles of NF-κB, STAT1 and IRF1 in IAA-induced IDO1 expression (**D**). Their TCMs were used to treat CD8^+^T cells to examine their impacts on viability and the proliferation of CD8^+^T cells (**E**). The IDO1 concentration in those TCMs was examined (**F**). The Trp concentration (**G**) and [Kyn]/[Trp] ratio (**H**) in the media of TCM-treated CD8^+^T cells, as well as their cell viability (**I**), proliferation (**J**), and apoptosis (**K**) were examined. For in vivo exams, shVector cells, shVector cells with PDTC treatment (50 mg/kg), shSTAT1 cells, shSTAT1 cells with PDTC treatment (50 mg/kg), and shIRF1 cells were inoculated in C57BL/6 mice, and the tumor grafts were treated with the CD8^+^T conditional medium as well as PDTC (**L**) (each group n = 10). Tumor growth was examined (**M**) (“*” indicates the difference between VPCT and SVCT; “#” indicates the difference between STCT and SVCT; “+” indicates the difference between IFCT and SVCT). Tumor grafts and blood were collected for the analyses of IDO1 expression (**N**), the relative Trp content (**O**) and [Kyn]/[Trp] ratio (**P**), and relative proportions of CD8^+^T cells (**Q**). “*” or “#” indicates *p* < 0.05; “**”, “##”, or “++” indicates *p* < 0.01; “***” or “+++” indicates *p* < 0.001.

**Figure 7 biomedicines-12-00573-f007:**
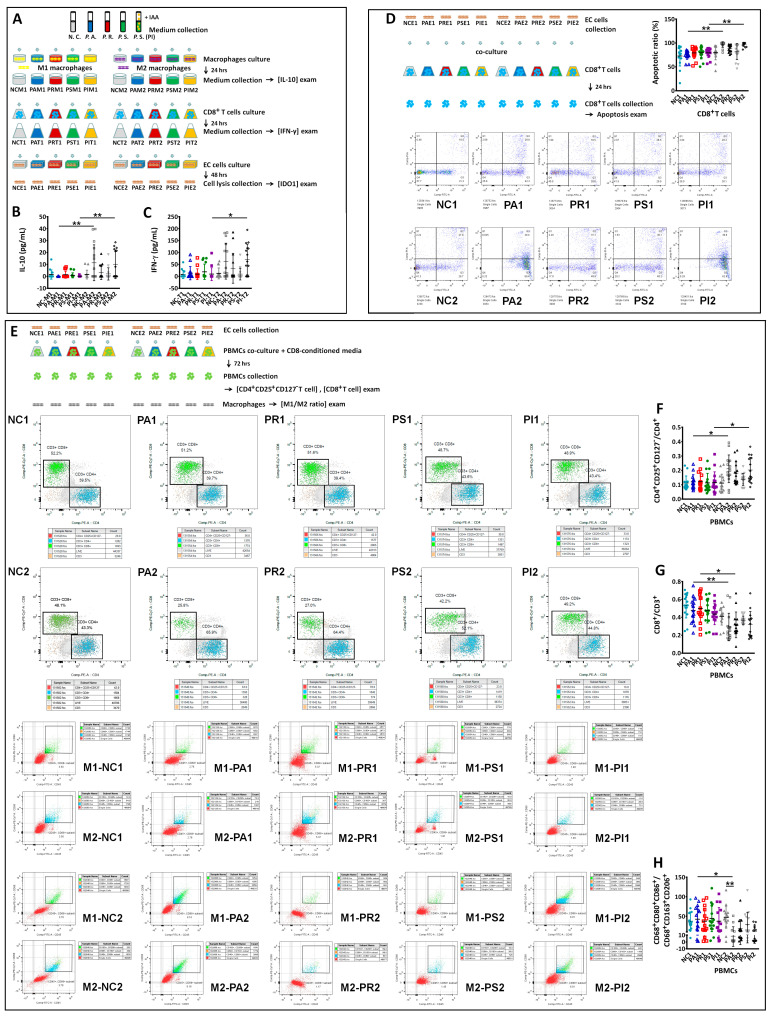
Scheme of the experiments (**A**) to verify which macrophages, M1 or M2, produced IL-10 after *Peptostreptococcus* species stimulation. The examination of IL-10 (**B**) and IFN-γ (**C**), as described in (**A**). Conditioned medium-treated EC cells, as described in panel A, were co-cultured with CD8^+^T cells for 24 h, and the apoptosis of CD8^+^T cells was examined (**D**). The scheme of the experiments (**E**) was to investigate the effect of EC cells on T_reg_ and CD8^+^T cell proportions and the M1/M2 macrophage ratio in PBMCs. The examination of CD4^+^CD25^+^CD127^−^T cell proportion (**F**), CD8^+^T cell proportion (**G**), and the ratio of M1/M2 (**H**), as described in (**E**). “*” indicates *p* < 0.05; “**” indicates *p* < 0.01.

**Figure 8 biomedicines-12-00573-f008:**
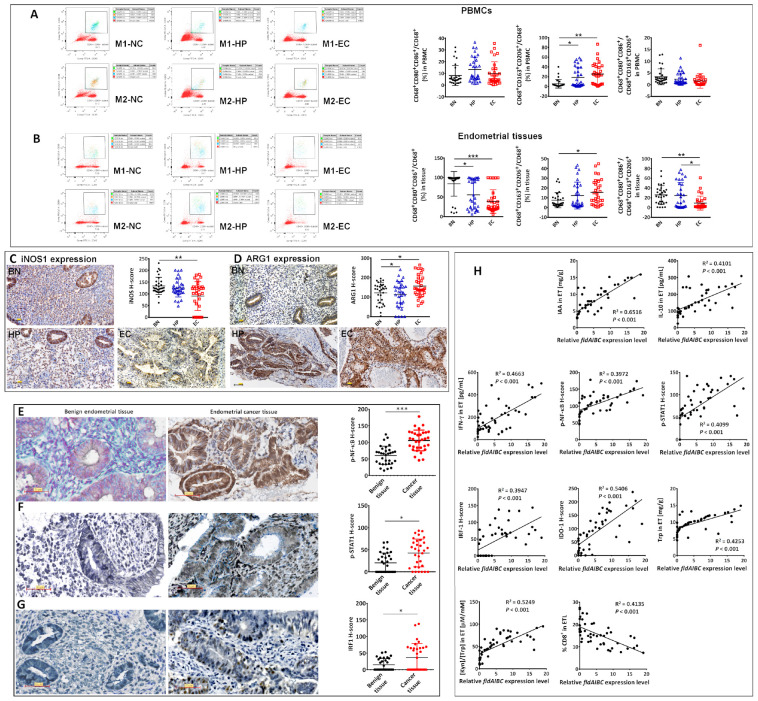
Clinical examination of M1/M2 ratio in PBMCs (**A**) and the endometrial tissue (**B**) of BN, HP, and EC patients. iNOS and ARG1 expression (**C**,**D**) in the endometrial tissue of BN, HP, and EC patients. Expressions of p-NF-κB, p-STAT1, and IRF1 were examined in EC tissues (n = 32) and benign endometrial tissues (n = 32) (**E**–**G**). Linear regression was used to test the fitting degrees between *fldAIBC* expression and the following clinical values (**H**), including the IAA, IL-10, IFN-γ and Trp content, [Kyn]/[Trp] ratio, expressions of p-NF-κB, p-STAT1, IRF1 and IDO1, as well as CD8^+^T proportions. “*” indicates *p* < 0.05; “**” indicates *p* < 0.01; “***” indicates *p* < 0.001. Scale bar: 50 µm (**C**,**D**).

**Figure 9 biomedicines-12-00573-f009:**
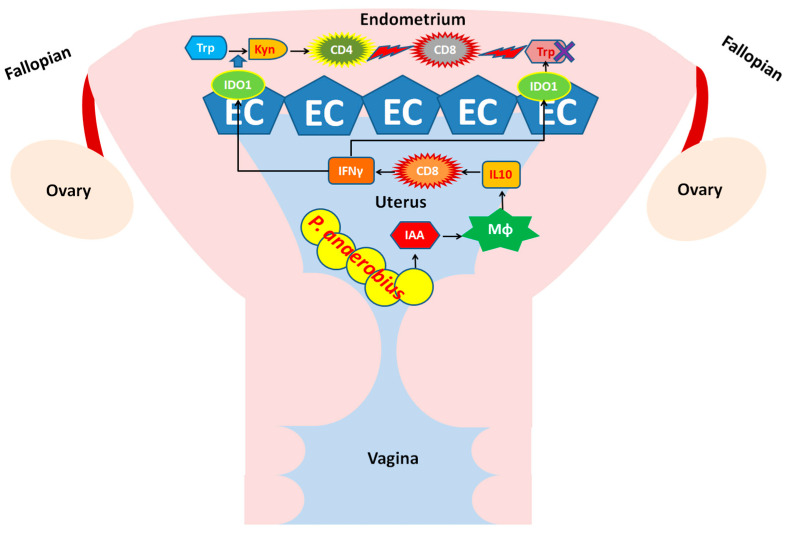
Illustration of the contribution of uterine commensal *P.* anaerobius to IDO1 expression and the impact on infiltrated lymphocytes in the endometrium of EC.

**Table 1 biomedicines-12-00573-t001:** Numbers, ages, BMI, and menopausal status of BN, HP, and EC patients, including EC stages and grades, and types of HP and BN.

	EC	HP	BN	*p* Value		
				EC vs. HP	EC vs. BN	HP vs. BN
n	32	32	32			
Age	53 ± 11	50 ± 13	50 ± 8	>0.05	>0.05	>0.05
Years, mean ± SD						
BMI	23.9 ± 3.4	23.1 ± 2.9	24.4 ± 3.9	>0.05	>0.05	>0.05
kg/m^2^, mean ± SD						
Menopausal status						
Post-menopause n (%)	26 (81%)	22 (69%)	26 (81%)	>0.05	>0.05	>0.05
Pre-menopause n (%)	6 (19%)	10 (31%)	6 (19%)			
FIGO stage						
n (%)						
I	27 (84)	-	-			
II	1 (3)	-	-			
III	4 (13)	-	-			
Grade, n (%)						
1	15 (47)	-	-			
2	14 (44)	-	-			
3	3 (9)	-	-			
Hyperplasia						
Simple n (%)	-	29 (91)	-			
Complex n (%)	-	1 (3)	-			
Atypical n (%)	-	2 (6)	-			
n (%)						
Hysteromyoma	-	-	29 (91%)			
Benign ovarian cyst	-	-	3 (9%)			

## Data Availability

Data available on request from the authors.

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
