# Peer review of "Uterine Commensal Peptostreptococcus Species Contribute to IDO1 Induction in Endometrial Cancer via Indoleacrylic Acid"

_biomedicines, 2024, doi:10.3390/biomedicines12030573_

Round 1

Reviewer 1 Report

Comments and Suggestions for Authors

This is an interesting study

1. The main question addressed by the research is if Peptostreptococcus species are related to Endometrial Cancer.

2. The topic is relevant to the field and partially original. However, the coexistence of Peptostreptococcus Species in cases of Endometrial Cancer does not necessarily means “causality”

3. Compared to other published material, the manuscript focuses on the special relation of Peptostreptococcus Species with Endometrial Cancer.

4. As already mentioned, the coexistence of Peptostreptococcus Species in cases of Endometrial Cancer does not necessarily means “causality” and, thus, sentences like “This work highlights the role of microbial dysbiosis in carcinogenesis…” should be altered. 

5. The conclusions are consistent with the evidence and arguments presented. However, a prospective study with adequate number of patients (including those with endometrial cancer without the coexistence of Peptostreptococcus Species) could add specific information to the conclusions.

6. The references are appropriate. However, some important references related to the subject are missing. Example: 

Supratim Mandal S, et al. Human microbial dysbiosis as driver of gynecological malignancies. Review  Biochimie 2022;197:86-95.  

Author Response

This is an interesting study
1. The main question addressed by the research is if Peptostreptococcus species are related to Endometrial Cancer.
Re: yes, that is.

2. The topic is relevant to the field and partially original. However, the coexistence of Peptostreptococcus Species in cases of Endometrial Cancer does not necessarily means “causality”.
Re: we fully agree with your opinion. However, our results suggest that it has potential possibilities. We can confirm that: 
(1). The vaginal microbiome composition of women with gynaecological cancer is consistent with that of women with symptomatic bacterial vaginosis
(2). Gastrointestinal commensal Peptostreptococcus species produce a tryptophan metabolite, indoleacrylic acid (IAA) which suppresses inflammatory responses by promoting IL-10 expression in macrophages.
(3). IAA is one of the most abundant metabolites in the sera of type I endometrial cancer (EC) patients.
(4). Commensal Peptostreptococcus species exist in the genital tracts of healthy women of reproductive age, and mainly in their endometria, cervical mucus and posterior fornix.
(5). In EC patients, high IDO1 expression correlates with chemoresistance and low progression-free survival. And the IDO 1 expression in epithelial cells, is associated with the stimulation of IFN- γ.
According to our results, 
(1). the carrying rate of P. anaerobius flora of EC patients in endometria and posterior fornix was significantly higher than that of benign women in corresponding tissues and organs (Fig. 4D); 
(2). gene cluster fldAIBC that could induce IAA expression, its expression level in the endometria of EC patients was significantly higher than that in benign subjects (Fig. 4E).
Based on the above 5 published results and our results, we made and verified the following hypothesis:
Female genital tract - Peptostreptococcus species - IAA - IL-10 (macrophages) - IFN-γ (CD8T) - IDO1 (endometrial epithelium) - tryptophan degradation, Treg cells differentiation - Teff cells anergy.
There are many pathways to constitute this hypothesis, and although we can prove the possibility of each pathway, we can not prove the necessity of its occurrence. Therefore, we can only conclude that there is a possible causal relationship between the coexistence of Peptostreptococcus Species and the occurrence of endometrial cancer.

3. Compared to other published material, the manuscript focuses on the special relation of Peptostreptococcus Species with Endometrial Cancer.
Re: yes, it does. From the perspective of the occurrence and development process of chronic diseases, the long-term imbalance of local microflora may play a certain role in the occurrence of cancer.

4. As already mentioned, the coexistence of Peptostreptococcus Species in cases of Endometrial Cancer does not necessarily means “causality” and, thus, sentences like “This work highlights the role of microbial dysbiosis in carcinogenesis…” should be altered. 
Re: yes, thanks. 
What the above statement would suggest was that, this study highlighted the role of microbial dysbiosis in cancer development. Here, cancer does not mean endometrial cancer, but means kinds of cancer. The correlation of microbial dysbiosis and cancer occurrence has been established in some cancers, such as gastric cancer. Nevertheless, we still made modification in that sentence, like "This work highlights the possible causal relationship between microbial dysbiosis and carcinogenesis.".

5. The conclusions are consistent with the evidence and arguments presented. However, a prospective study with adequate number of patients (including those with endometrial cancer without the coexistence of Peptostreptococcus Species) could add specific information to the conclusions.
Re:in this study, we set three groups, namely, endometrial cancer group, hyperplasia group, and benign group, with 32 patients in each group. The enrollment was indeed quite limited. However, we still conducted the correlation analysis of microflora carriage and pathogenesis, including the correlation of Peptostreptococcus species in endometrium, cervical canal, posterior fornix of every patients and pathogenesis of endometrial cancer (Fig. 4). Sufficient enrollment would certainly make the findings more credible. However, we can still be certain that the pathogenesis of endometrial cancer is multifaceted, and the dysbiosis of Peptostreptococcus species may be only one of the reasons. 

6. The references are appropriate. However, some important references related to the subject are missing. Example: 
Supratim Mandal S, et al. Human microbial dysbiosis as driver of gynecological malignancies. Review  Biochimie 2022;197:86-95.  
Re: yes, thanks. It is appropriate to cite this publication. We have cited it. 

Reviewer 2 Report

Comments and Suggestions for Authors

Comments to the authors:

Wang et al. explore the role of uterine Peptostreptococcus dysbiosis in gynaecological cancer via indoleacrylic acid-induced indoleamine-2,3-dioxygenase 1 induction. Overall, the idea sounds interesting. So, I would like to suggest some modifications to improve its quality:

In a healthy state, the vaginal microbiota is regulated by estrogen levels. Also, estrogen might mediate cancer. These could be noted in line 54.

The dysbiosis of the vaginal microbiome not only contributes to gynaecological cancers but also increases the risk of acquiring sexually transmitted infections, preterm birth, spontaneous miscarriage, or pelvic inflammatory disease. This could be added to line 57.

In addition to uterine carcinogenesis, the vaginal microbiome contributed to cervical cancer and possibly other gynaecological cancers; the authors should mention it in the introduction.

In the introduction, the authors should expand on the direct and indirect mechanisms of microbially driven carcinogenesis, particularly gynaecological cancers.

Cancer therapies change the microbiota of the uterus. Cancer treatments can also be affected by the composition of the microbiota. These could be noted in the introduction.

In the method, the patient's age range was from 29 to 73 years. The vaginal microbiota might be influenced by the age and hormonal status of patients (I mean premenopausal versus postmenopausal patients). This might affect the data. The authors should explain this confounding variable.

All chemicals and reagents' catalogue numbers and devices' manufacturers should be noted in the entire methodology.

In the results, the expression of proteins in the representative figures should be shown with arrows or explained in the figure caption.

Author Response

Comments to the authors:
Wang et al. explore the role of uterine Peptostreptococcus dysbiosis in gynaecological cancer via indoleacrylic acid-induced indoleamine-2,3-dioxygenase 1 induction. Overall, the idea sounds interesting. So, I would like to suggest some modifications to improve its quality:

1. In a healthy state, the vaginal microbiota is regulated by estrogen levels. Also, estrogen might mediate cancer. These could be noted in line 54.
Re: Yes, thanks. We agree. Estrogen deficiency was proved to lead to lactobacillus loss, while vaginal lactobacillus levels were siginificantly increased by estrogen therapy in clinical trials. This indicates that estrogen plays a role in vaginal microbiota regulation and thereby might be involved in cancer mediation.

2. The dysbiosis of the vaginal microbiome not only contributes to gynaecological cancers but also increases the risk of acquiring sexually transmitted infections, preterm birth, spontaneous miscarriage, or pelvic inflammatory disease. This could be added to line 57.
Re: yes, thanks. We have added it.

3. In addition to uterine carcinogenesis, the vaginal microbiome contributed to cervical cancer and possibly other gynaecological cancers; the authors should mention it in the introduction.
Re: thanks. We simply described it briefly. The common gynecological tumors have cervical cancer, ovarian cancer, uterine body cancer. Cervical and ovarian cancer were not within the subject of our study. The purpose of this manuscript is to publish original findings, not to write a review. If the introduction discussion is too wide, it is not conducive to the expression of the main content of the current study.

4. In the introduction, the authors should expand on the direct and indirect mechanisms of microbially driven carcinogenesis, particularly gynaecological cancers.
Re: thanks. We have explained this point in paragraph 3 of the introduction.

5. Cancer therapies change the microbiota of the uterus. Cancer treatments can also be affected by the composition of the microbiota. These could be noted in the introduction.
Re: thanks for your advice. Those could be noted. This manuscript discusses a potential occurrence mechanism of type I endometrial cancer. This mechanism may be related to the dysbiosis in the uterus. In this manuscript, cancer treatment was not within the scope of our study. Therefore, we did not discuss the mechanism or the original research that were related to the treatment.

6. In the method, the patient's age range was from 29 to 73 years. The vaginal microbiota might be influenced by the age and hormonal status of patients (I mean premenopausal versus postmenopausal patients). This might affect the data. The authors should explain this confounding variable.
Re: thanks. You posed up a good question. According to our previous metabolomics study, IAA, the metabolite of Peptostreptococcus species, was highly expressed in EC patients and did not show an age correlation. That is to say, some endometrial cancer patients with high IAA expression were premenopausal women while some patients postmenopausal women. Meanwhile, in our analysis of genital tract bacterial samples, the differential distribution of Peptostreptococcus species was only between endometrial cancer patients and hyperplasia or benign patients. In the range of cancer patients, it did not show age correlation. We guess that if postmenopausal women, the estrogen level in the body is low, accordingly prone to microflora disorders. If a premenopausal woman, there is still a possibility of low estrogen levels for some reason. Therefore, it may promote the dysregulation of the distribution of microflora in their genital tract.

7. All chemicals and reagents' catalogue numbers and devices' manufacturers should be noted in the entire methodology.
Re: thanks for the suggestion. The work began from 2017, and the catalogue numbers of chemical reagents are not available. However, we describe the name of the reagent manufacturer and the location of the manufacturer in the main text. In addition, the manufacturers of the equipments used for analyses and testings are also expressed in the text.

8. In the results, the expression of proteins in the representative figures should be shown with arrows or explained in the figure caption.
Re: thanks for the suggestion. For the morphological detection of the expression of the target proteins, we used both immunohistochemistry and immunofluorescence. In each method, we explained the name of the target protein in the figure captions. In immunofluorescence, the carrying fluorescence of the target protein was also annotated with the text of the corresponding color in the figure.

Reviewer 3 Report

Comments and Suggestions for Authors

This study used endometrial cancer (EC) cell line, tumor-grafted mice, and clinical data from EC patients to explore the production of indoleacrylic acid (IAA) and through indoleamine-2,3-dioxygenase 1 (IDO1) expression by the female genital tract Peptostreptococcus species, and its effect on Treg and Teff cells, M1 and M2 macrophages population, to understand the relationship between IAA, IDO1 and Peptostreptococcus species on type I EC. This is a rigorous research paper. The authors designed detailed and complete experiments for the many hypotheses they proposed, and provided a lot of meaningful and valuable experimental data. Moreover, they are also very attentive and careful in preparing data charts. Taking together, this paper deserves to be published in this journal.

Specific comments

1.       Figure 1-8 provide a lot of experimental data, but it is squeezed into a small space and difficult to read. I suggest that these eight pictures be enlarged or separated into more figures.

2.       Fig. 2G: The names of the two photos are not visible on the black and white paper, I recommend changing the colors or placing the names outside the photos.

3.       Fig. 8H: The R2 is generally used in data regression to see the differences between the regression line and the data, so it is called the "determination coefficient". In this figure, the R value, which is called the "correlation coefficient", should be used to test the correlation between the two parameters on the x and y axes.

4.       Line 272-273: Two paragraphs should be connected.

5.       Line 344: The name of city and country where Bio-Rad is located should be moved to line 340.

6.       Line 713, and line 853-854: Microorganism names should be in italics.

Author Response

This study used endometrial cancer (EC) cell line, tumor-grafted mice, and clinical data from EC patients to explore the production of indoleacrylic acid (IAA) and through indoleamine-2,3-dioxygenase 1 (IDO1) expression by the female genital tract Peptostreptococcus species, and its effect on Treg and Teff cells, M1 and M2 macrophages population, to understand the relationship between IAA, IDO1 and Peptostreptococcus species on type I EC. This is a rigorous research paper. The authors designed detailed and complete experiments for the many hypotheses they proposed, and provided a lot of meaningful and valuable experimental data. Moreover, they are also very attentive and careful in preparing data charts. Taking together, this paper deserves to be published in this journal.

Specific comments

1. Figure 1-8 provide a lot of experimental data, but it is squeezed into a small space and difficult to read. I suggest that these eight pictures be enlarged or separated into more figures.
Re: thanks. In the PDF version used for the review, all pictures were integrated into the PDF file. Each big picture can be seen clearly, but the small images in the big pictures are hard to clearly see. In fact, in our submitted manuscript, each figure was submitted in TIFF picture format. In each large figure submitted in TIFF picture format, you can zoom in at the small images in each one and you can read clearly.

2. Fig. 2G: The names of the two photos are not visible on the black and white paper, I recommend changing the colors or placing the names outside the photos.
Re: thanks. The names of the two images in Figure 2G can be seen. Nowadays the pictures are color pictures: you can see the picture online on the Internet, you can also be printed out to see the picture, and the name of the picture can be seen. If the printed image is in black and white, the image name can also be distinguished. At the same time, combined with the legend and online reading the color pictures, the name of the figure can be recognized.

3. Fig. 8H: The R2 is generally used in data regression to see the differences between the regression line and the data, so it is called the "determination coefficient". In this figure, the R value, which is called the "correlation coefficient", should be used to test the correlation between the two parameters on the x and y axes.
Re: thanks for your correction. That is kind suggestion. In fact, Linear regression was used to test the fitting degree between the expression of fldAIBC and the following clinical values. We have corrected them accordingly. 

4. Line 272-273: Two paragraphs should be connected.
Re: Line 272-273, the two paragraphs belong to different paragraphs and need to be separated into two paragraphs. 

5. Line 344: The name of city and country where Bio-Rad is located should be moved to line 340.
Re: thanks for the suggestion. We have corrected it accordingly.

6. Line 713, and line 853-854: Microorganism names should be in italics.
Re: thanks for the suggestion. We have corrected it accordingly.

Round 2

Reviewer 1 Report

Comments and Suggestions for Authors

Example of alterations: In the original text, in line 377, it was written “IFN-γ treatment induced IDO1 in Ishikawa endometrial adenocarcinoma cells (Fig. 2C).” giving the impression of results related to the “Materials…” of the study. However, according to related correspondence, “The authors could not find original images of WB and removed them as a result”. Thus, among others, you should clearly support (and “declare”) the reliability of your data and related findings.

Author Response

Comments and Suggestions for Authors

Example of alterations: In the original text, in line 377, it was written “IFN-γ treatment induced IDO1 in Ishikawa endometrial adenocarcinoma cells (Fig. 2C).” giving the impression of results related to the “Materials…” of the study. However, according to related correspondence, “The authors could not find original images of WB and removed them as a result”. Thus, among others, you should clearly support (and “declare”) the reliability of your data and related findings.

Re: thanks for your comments and suggestions.

This study was done about 5 years ago. Then, we tried to submit it to some journals as well. However, this manuscript met several difficulties including drafting and submission. From dysbiosis in female genital tract to IDO1 expressed by endometrial cancer cells and subsequent CD8T cells anergy, many mechanisms are involved, resulting to the content of the text obscure to understand. Moreover, each mechanism linked has its own probability of occurrence. Overall, according to the conclusion of this manuscript, the probability of IDO 1 expression in endometrial cells according to this mechanism could be low. Therefore, our articles were rejected several times by other journals. Over the past five years, there have been personnel changes and the handover of data results. We did not carefully examine the data for each trial during the previous data handover. Therefore, some original data in this study cannot be found, such as western blotting film and original photos. As the missing of the original data, we have removed the western blotting methods and results from the manuscript as suggested by the editors and reviewers.

Reviewer 2 Report

Comments and Suggestions for Authors

Thanks to the authors for their effort. 

Author Response

Comments and Suggestions for Authors

Thanks to the authors for their effort.

Re: thanks for your comments and suggestions.

Reviewer 3 Report

Comments and Suggestions for Authors

1.      This paper has been revised in accordance with my previous comments.

2.      Regarding the issue raised by the Editor about the authors being unable to provide the original images of Western blots part and deleting the related photos, after reviewing the relevant content, I think that it does not greatly affect the presentation of the results of this paper.

3.      In summary, this revised version of article is acceptable for publication in this journal.

Author Response

Comments and Suggestions for Authors

  1. This paper has been revised in accordance with my previous comments.
  2. Regarding the issue raised by the Editor about the authors being unable to provide the original images of Western blots part and deleting the related photos, after reviewing the relevant content, I think that it does not greatly affect the presentation of the results of this paper.
  3. In summary, this revised version of article is acceptable for publication in this journal.

Re: thanks for your comments and suggestions.